# A Unified View of Label Shift Estimation

**Saurabh Garg, Yifan Wu, Sivaraman Balakrishnan, Zachary C. Lipton**
Machine Learning Department,
Department of Statistics and Data Science,
Carnegie Mellon University
{sgarg2,yw4,sbalakri,zlipton}@andrew.cmu.edu

## Abstract

Under label shift, the label distribution $p(y)$ might change but the class-conditional distributions $p(x|y)$ do not. There are two dominant approaches for estimating the label marginal. BBSE, a moment-matching approach based on confusion matrices, is provably consistent and provides interpretable error bounds. However, a maximum likelihood estimation approach, which we call MLLS, dominates empirically. In this paper, we present a unified view of the two methods and the first theoretical characterization of MLLS. Our contributions include (i) consistency conditions for MLLS, which include calibration of the classifier and a confusion matrix invertibility condition that BBSE also requires; (ii) a unified framework, casting BBSE as roughly equivalent to MLLS for a particular choice of calibration method; and (iii) a decomposition of MLLS's finite-sample error into terms reflecting miscalibration and estimation error. Our analysis attributes BBSE's statistical inefficiency to a loss of information due to coarse calibration. Experiments on synthetic data, MNIST, and CIFAR10 support our findings.

## 1   Introduction

Supervised algorithms are typically developed and evaluated assuming independent and identically distributed (iid) data. However, the real world seldom abides, presenting domain adaptation problems in which the *source distribution* $\mathrm{P}_s$, from which we sample labeled training examples, differs from the *target distribution* $\mathrm{P}_t$, from which we only observe unlabeled data. Absent assumptions on the nature of shift, the problem is underspecified. Multiple assumptions may be compatible with the same observations while implying different courses of action. Fortunately, some assumptions can render shift detection, estimation, and on-the-fly updates to our classifiers possible.

This paper focuses on *label shift* [22, 18, 16], which aligns with the *anticausal* setting in which the labels $y$ cause the features $x$ [19]. Label shift arises in diagnostic problems because diseases cause symptoms. In this setting, an intervention on $p(y)$ induces the shift, but the process generating $x$ given $y$ is fixed ($p_s(x|y) = p_t(x|y)$). Under label shift, the optimal predictor may change, e.g., the probability that a patient suffers from a disease given their symptoms can increase under a pandemic. Contrast label shift with the better-known *covariate shift* assumption, which aligns with the assumption that $x$ causes $y$, yielding the reverse implication that $p_s(y|x) = p_t(y|x)$.

Under label shift, our first task is to estimate the ratios $w(y) = p_t(y)/p_s(y)$ for all labels $y$. Two dominant approaches leverage off-the-shelf classifiers to estimate $w$: (i) *Black Box Shift Estimation* (BBSE) [16] and a variant called *Regularized Learning under Label Shift* (RLLS) [2]: moment-matching based estimators that leverage (possibly biased, uncalibrated, or inaccurate) predictions to estimate the shift; and (ii) Maximum Likelihood Label Shift (MLLS) [18]: an Expectation Maximization (EM) algorithm that assumes access to a classifier that outputs the true source distribution conditional probabilities $p_s(y|x)$.

Given a predictor $\widehat{f}$ with an invertible confusion matrix, BBSE and RLLS have known consistency results and finite-sample guarantees [16, 2]. However, MLLS, in combination with a calibration heuristic called Bias-Corrected Temperature Scaling (BCTS), outperforms them empirically [1].

In this paper, we theoretically characterize MLLS, establishing conditions for consistency and bounding its finite-sample error. To start, we observe that given the true label conditional $p_s(y|x)$, MLLS is simply a concave Maximum Likelihood Estimation (MLE) problem and standard results apply. However, because we never know $p_s(y|x)$ exactly, MLLS is always applied with an estimated model $\widehat{f}$ and thus the procedure consists of MLE under model misspecification.

First, we prove that (i) *canonical calibration* (Definition 1) and (ii) an invertible confusion matrix (as required by BBSE) are *sufficient conditions* to ensure MLLS's consistency (Proposition 1, Theorems 1 and 2). We also show that calibration can sometimes be *necessary* for consistency (Example 1 in Section 4.3). Recall that neural network classifiers tend to be uncalibrated absent post-hoc adjustments [10]. Second, we observe that confusion matrices can be instruments for calibrating a classifier. Applying MLLS with this technique, BBSE and MLLS are distinguished only by their objective functions. Through extensive experiments, we show that they perform similarly, concluding that MLLS's superior performance (when applied with more granular calibration techniques) is not due to its objective but rather to the information lost by BBSE via confusion matrix calibration. Third, we analyze the finite-sample error of the MLLS estimator by decomposing its error into terms reflecting the miscalibration error and finite-sample error (Theorem 3). Depending on the calibration method, the miscalibration error can further be divided into two terms: finite sample error due to re-calibration on a validation set and the minimum achievable calibration error with that technique.

We validate our results on synthetic data, MNIST, and CIFAR-10. Empirical results show that MLLS can have 2–10$\times$ lower Mean Squared estimation Error (MSE) depending on the magnitude of the shift. Our experiments relate MLLS's MSE to the granularity of the calibration.

In summary, we contribute the following: (i) Sufficient conditions for MLLS's consistency; (ii) Unification of MLLS and BBSE methods under a common framework, with BBSE corresponding to a particular choice of calibration method; (iii) Finite-sample error bounds for MLLS; (iv) Experiments on synthetic and image recognition datasets that support our theoretical arguments.

## 2 Problem Setup

Let $\mathcal{X}$ be the input space and $\mathcal{Y} = \{1, 2, \ldots, k\}$ the output space. Let $\mathrm{P}_s, \mathrm{P}_t : \mathcal{X} \times \mathcal{Y} \to [0, 1]$ be the source and target distributions and let $p_s$ and $p_t$ denote the corresponding probability density (or mass) functions. We use $\mathbb{E}_s$ and $\mathbb{E}_t$ to denote expectations over the source and target distributions. In unsupervised domain adaptation, we possess labeled source data $\{(x_1, y_1), (x_2, y_2), \ldots, (x_n, y_n)\}$ and unlabeled target data $\{x_{n+1}, x_{n+2}, \ldots, x_{n+m}\}$. We also assume access to a black-box predictor $\widehat{f} : \mathcal{X} \mapsto \Delta^{k-1}$, e.g., a model trained to approximate the true probability function $f^*$, where $f^*(x) := p_s(\cdot|x)$. Here and in the rest of the paper, we use $\Delta^{k-1}$ to denote the standard $k$-dimensional probability simplex. For a vector $v$, we use $v_y$ to access the element at index $y$.

Absent assumptions relating the source and target distributions, domain adaptation is underspecified [3]. We work with the *label shift* assumption, i.e., $p_s(x|y) = p_t(x|y)$, focusing on multiclass classification. Moreover, we assume non-zero support for all labels in the source distribution: for all $y \in \mathcal{Y}, p_s(y) \geq c > 0$ [16, 2]. Under label shift, three common goals are (i) detection—determining whether distribution shift has occurred; (ii) quantification—estimating the target label distribution; and (iii) correction—producing a predictor that minimizes error on the target distribution [16].

This paper focuses on goal (ii), estimating importance weights $w(y) = p_t(y)/p_s(y)$ for all $y \in \mathcal{Y}$. Given $w$, we can update our classifiers on the fly, either by retraining in an importance-weighted ERM framework [20, 9, 16, 2]—a practice that may be problematic for overparameterized neural networks [5], or by applying an analytic correction [1, 18]. Within the ERM framework, the generalization result from Azizzadenesheli et al. [2] (Theorem 1) depends only on the error of the estimated weights, and hence any method that improves weight estimates tightens this bound.

There are multiple definitions of calibration in the multiclass setting. Guo et al. [10] study the calibration of the arg-max prediction, while Kumar et al. [14] study a notion of per-label calibration. We use canonical calibration [24] and the expected canonical calibration error on the source data defined as follows:

**Definition 1** (Canonical calibration). *A prediction model $f : \mathcal{X} \mapsto \Delta^{k-1}$ is canonically calibrated on the source domain if for all $x \in \mathcal{X}$ and $j \in \mathcal{Y}$, $\mathrm{P}_s(y = j | f(x)) = f_j(x)$.*

**Definition 2** (Expected canonical calibration error). *For a predictor $f$, the expected squared canonical calibration error on the source domain is $\mathcal{E}^2(f) = \mathbb{E}_s \|f - f_c\|^2$, where $f_c = \mathrm{P}_s(y = \cdot | f(x))$.*

Calibration methods typically work either by calibrating the model during training or by calibrating a trained classifier on held-out data, post-hoc. We refer the interested reader to Kumar et al. [14] and Guo et al. [10] for detailed studies on calibration. We focus on the latter category of methods. Our experiments follow Alexandari et al. [1], who leverage BCTS [1] to calibrate their models. BCTS extends temperature scaling [10] by incorporating per-class bias terms.

## 3  Prior Work

Two families of solutions have been explored that leverage a blackbox predictor: BBSE [16], a moment matching method, uses the predictor $\widehat{f}$ to compute a confusion matrix $C_{\widehat{f}} := p_s(\widehat{y}, y) \in \mathbb{R}^{k \times k}$ on the source data. Depending on how $\widehat{y}$ is defined, there are two types of confusion matrix for a predictor $\widehat{f}$: (i) the *hard confusion matrix* $\widehat{y} = \arg\max \widehat{f}(x)$; and (ii) the *soft confusion matrix*, where $\widehat{y}$ is defined as a random prediction that follows the discrete distribution $\widehat{f}(x)$ over $\mathcal{Y}$. Both soft and hard confusion matrix can be estimated from labeled source data samples. The estimate $\widehat{w}$ is computed as $\widehat{w} := \widehat{C}_{\widehat{f}}^{-1} \widehat{\mu}$, where $\widehat{C}_{\widehat{f}}$ is the estimate of confusion matrix and $\widehat{\mu}$ is an estimate of $p_t(\widehat{y})$, computed by applying the predictor $\widehat{f}$ to the target data. In a related vein, RLLS [2] incorporates an additional regularization term of the form $\|w - 1\|$ and solves a constrained optimization problem to estimate the shift ratios $w$.

MLLS estimates $w$ as if performing maximum likelihood estimation, but substitutes the predictor outputs for the true probabilities $p_s(y|x)$. Saerens et al. [18], who introduce this procedure, describe it as an application of EM. However, as observed in [8, 1], the likelihood objective is concave, and thus a variety of optimization algorithms may be applied to recover the MLLS estimate. Alexandari et al. [1] also showed that MLLS underperforms BBSE when applied naively, a phenomenon that we shed more light on in this paper.

## 4  A Unified View of Label Shift Estimation with Black Box Predictors

We now present a unified view that subsumes MLLS and BBSE and demonstrate how each is instantiated under this framework. We also establish identifiability and consistency conditions for MLLS, deferring a treatment of finite-sample issues to Section 5. For convenience, throughout Sections 3 and 4, we use the term *calibration* exclusively to refer to canonical calibration (Definition 1) on the source data. We relegate all technical proofs to Appendix D.

### 4.1  A Unified Distribution Matching View

To start, we introduce a *generalized* distribution matching approach for estimating $w$. Under label shift, for any (possibly randomized) mapping from $\mathcal{X}$ to $\mathcal{Z}$, we have that $p_s(z|y) = p_t(z|y)$ since, $p_s(z|y) = p_t(z|y) = \int_{\mathcal{X}} p(z|x)p(x|y)dx$. Throughout the paper, we use the notation $p(z|y)$ to represent either $p_s(z|y)$ or $p_t(z|y)$ (which are identical). We now define a family of distributions over $\mathcal{Z}$ parameterized by $w \in \mathcal{W}$ as

$$p_w(z) = \sum\nolimits_{y=1}^{k} p(z|y)p_s(y)w_y = \sum\nolimits_{y=1}^{k} p_s(z, y)w_y, \qquad (1)$$

where $\mathcal{W} = \{w \mid \forall y, w_y \geq 0 \text{ and } \sum_{y=1}^{k} w_y p_s(y) = 1\}$. When $w = w^*$, we have that $p_w(z) = p_t(z)$. For fixed $p(z|x)$, $p_t(z)$ and $p_s(z, y)$ are known because $p_t(x)$ and $p_s(x, y)$ are known. So one potential strategy to estimate $w^*$ is to find a weight vector $w$ such that

$$\sum\nolimits_{y=1}^{k} p_s(z, y)w_y = p_t(z) \quad \forall z \in \mathcal{Z}. \qquad (2)$$

At least one such weight vector $w$ must exist as $w^*$ satisfies (2). We now characterize conditions under which the weight vector $w$ satisfying (2) is unique:

**Lemma 1** (Identifiability). *If the set of distributions $\{p(z|y) : y = 1, ..., k\}$ are linearly independent, then for any $w$ that satisfies (2), we must have $w = w^*$. This condition is also necessary in general: if the linear independence does not hold then there exists a problem instance where we have $w, w^* \in \mathcal{W}$ satisfying (2) while $w \neq w^*$.*

Lemma 1 follows from the fact that (2) is a linear system with at least one solution $w^*$. This solution is unique when $p_s(z, y)$ is of rank $k$. The linear independence condition in Lemma 1, in general, is sufficient for identifiability of discrete $\mathcal{Z}$. However, for continuous $\mathcal{Z}$, the linear dependence condition has the undesirable property of being sensitive to changes on sets of measure zero. By changing a collection of linearly dependent distributions on a set of measure zero, we can make them linearly independent. As a consequence, we impose a *stronger* notion of identifiability i.e., the set of distributions $\{p(z|y) : y = 1, ..., k\}$ are such that there does not exist $v \neq 0$ for which $\int_{\mathcal{Z}} |\sum_y p(z|y)v_y| dz = 0$. We refer this condition as *strict linear independence*.

In generalized distribution matching, one can set $p(z|x)$ to be the Dirac delta function at $\delta_x$[2] such that $\mathcal{Z}$ is the same space as $\mathcal{X}$, which leads to solving (2) with $z$ replaced by $x$. In practice where $\mathcal{X}$ is high-dimensional and/or continuous, approximating the solution to (2) from finite samples can be hard when choosing $z = x$. Our motivation for generalizing distribution matching from $\mathcal{X}$ to $\mathcal{Z}$ is that the solution to (2) can be better approximated using finite samples when $\mathcal{Z}$ is chosen carefully. Under this framework, the design of a label shift estimation algorithm can be decomposed into two parts: (i) the choice of $p(z|x)$ and (ii) how to approximate the solution to (2). Later on, we consider how these design choices may affect label shift estimation procedures in practice.

## 4.2 The Confusion Matrix Approach

If $\mathcal{Z}$ is a discrete space, one can first estimate $p_s(z, y) \in \mathbb{R}^{|\mathcal{Z}| \times k}$ and $p_t(z) \in \mathbb{R}$, and then subsequently attempt to solve (2). Confusion matrix approaches use $\mathcal{Z} = \mathcal{Y}$, and construct $p(z|x)$ using a black box predictor $\hat{f}$. There are two common choices to construct the confusion matrix: (i) The soft confusion matrix approach: We set $p(z|x) := \hat{f}(x) \in \Delta^{k-1}$. We then define a random variable $\hat{y} \sim \hat{f}(x)$ for each $x$. Then we construct $p_s(z, y) = p_s(\hat{y}, y)$ and $p_t(z) = p_t(\hat{y})$. (ii) The hard confusion matrix approach: Here we set $p(z|x) = \delta_{\arg\max \hat{f}(x)}$. We then define a random variable $\hat{y} = \arg\max \hat{f}(x)$ for each $x$. Then again we have $p_s(z, y) = p_s(\hat{y}, y)$ and $p_t(z) = p_t(\hat{y})$.

Since $p_s(z, y)$ is a square matrix, the identifiability condition becomes the invertibility of the confusion matrix. Given an estimated confusion matrix, one can find $w$ by inverting the confusion matrix (BBSE) or minimizing some distance between the vectors on the two sides of (2).

## 4.3 Maximum Likelihood Label Shift Estimation

When $\mathcal{Z}$ is a continuous space, the set of equations in (2) indexed by $\mathcal{Z}$ is intractable. In this case, one possibility is to find a weight vector $\widetilde{w}$ by minimizing the KL-divergence $\mathrm{KL}(p_t(z), p_w(z)) = \mathbb{E}_t [\log p_t(z)/p_w(z)]$, for $p_w$ defined in (1). This is equivalent to maximizing the population log-likelihood: $\widetilde{w} := \arg\max_{w \in \mathcal{W}} \mathbb{E}_t [\log p_w(z)]$. One can further show that $\mathbb{E}_t [\log p_w(z)] = \mathbb{E}_t[\log \sum_{y=1}^k p_s(z, y)w_y] = \mathbb{E}_t[\log \sum_{y=1}^k p_s(y|z)p_s(z)w_y] = \mathbb{E}_t[\log \sum_{y=1}^k p_s(y|z)w_y] + \mathbb{E}_t [\log p_s(z)]$. Therefore we can equivalently define:

$$\widetilde{w} := \arg\max_{w \in \mathcal{W}} \mathbb{E}_t \Big[ \log \sum_{y=1}^k p_s(y|z)w_y \Big]. \tag{3}$$

This yields a straightforward convex optimization problem whose objective is bounded from below [1, 8]. Assuming access to labeled source data and unlabeled target data, one can maximize the empirical counterpart of the objective in (3), using either EM or an alternative iterative optimization scheme. Saerens et al. [18] derived an EM algorithm to maximize the objective (3) when $z = x$, assuming access to $p_s(y|x)$. Absent knowledge of the ground truth $p_s(y|x)$, we can plug in any approximate predictor $f$ and optimize the following objective:

$$w_f := \arg\max_{w \in \mathcal{W}} \mathcal{L}(w, f) := \arg\max_{w \in \mathcal{W}} \mathbb{E}_t \left[ \log f(x)^T w \right]. \tag{4}$$

In practice, $f$ is fit from a finite sample drawn from $p_s(x, y)$ and standard machine learning methods often produce uncalibrated predictors. While BBSE and RLLS are provably consistent whenever the

predictor $f$ yields an invertible confusion matrix, to our knowledge, no prior works have established sufficient conditions to guarantee MLLS' consistency when $f$ differs from $p_s(y|x)$.

It is intuitive that for some values of $f \neq p_s(y|x)$, MLLS will yield inconsistent estimates. Supplying empirical evidence, Alexandari et al. [1] show that MLLS performs poorly when $f$ is a vanilla neural network predictor learned from data. However, Alexandari et al. [1] also show that in combination with a particular post-hoc calibration technique, MLLS achieves low error, significantly outperforming BBSE and RLLS. As the calibration error is not a distance metric between $f$ and $p_s(y|x)$ (zero calibration error does not indicate $f = p_s(y|x)$), a calibrated predictor $f$ may still be substantially different from $p_s(y|x)$. Some natural questions then arise:

1. *Why does calibration improve MLLS so dramatically?*
2. *Is calibration necessary or sufficient to ensure the consistency of MLLS?*
3. *What accounts for the comparative efficiency of MLLS over BBSE?* (Addressed in Section 5)

To address the first two questions, we make the following observations. Suppose we define $z$ (for each $x$) with distribution $p(z|x) := \delta_{f(x)}$, for some calibrated predictor $f$. Then, because $f$ is calibrated, it holds that $p_s(y|z) = f(x)$. Note that in general, the MLLS objective (4) can differ from (3). However, when $p(z|x) := \delta_{f(x)}$, the two objectives are identical. We can formalize this as follows:

**Lemma 2.** *If $f$ is calibrated, then the two objectives* (3) *and* (4) *are identical when $\mathcal{Z}$ is chosen as $\Delta^{k-1}$ and $p(z|x)$ is defined to be $\delta_{f(x)}$.*

Lemma 2 follows from changing the variable of expectation in (4) from $x$ to $f(x)$ and applying $f(x) = p_s(y|f(x))$ (definition of calibration). It shows that MLLS with a calibrated predictor on the input space $\mathcal{X}$ is in fact equivalent to performing distribution matching in the space $\mathcal{Z}$. Building on this observation, we now state our population-level consistency theorem for MLLS:

**Theorem 1** (Population consistency of MLLS). *If a predictor $f : \mathcal{X} \mapsto \Delta^{k-1}$ is calibrated and the distributions $\{p(f(x)|y) : y = 1, \ldots, k\}$ are strictly linearly independent, then $w^*$ is the unique maximizer of the MLLS objective* (4).

We now turn our attention to establishing consistency of the sample-based estimator. Let $x_1, x_2, \ldots, x_m \overset{iid}{\sim} p_t(x)$. The finite sample objective for MLLS can be written as

$$\widehat{w}_f := \arg\max_{w \in \mathcal{W}} \frac{1}{m} \sum_{i=1}^{m} \log f(x_i)^T w := \arg\max_{w \in \mathcal{W}} \mathcal{L}_m(w, f). \tag{5}$$

**Theorem 2** (Consistency of MLLS). *If $f$ satisfies the conditions in Theorem 1, then $\widehat{w}_f$ in* (5) *converges to $w^*$ almost surely.*

The main idea of the proof of Theorem 2 is to derive a metric entropy bound on the class of functions $\mathcal{G} = \left\{ (f^T w)/(f^T w + f^T w^*) | w \in \mathcal{W} \right\}$ to prove Hellinger consistency (Theorem 4.6 [25]). The consistency of MLLS relies on the linear independence of the collection of distributions $\{p(f(x)|y) : y = 1, \ldots, k\}$. The following result develops several alternative equivalent characterizations of this linear independence condition.

**Proposition 1.** *For a calibrated predictor $f$, the following statements are equivalent:*

*(1) $\{p(f(x)|y) : y = 1, \ldots, k\}$ are strictly linearly independent.*
*(2) $\mathbb{E}_s \left[ f(x)f(x)^T \right]$ is invertible.*
*(3) The soft confusion matrix of $f$ is invertible.*

Proposition 1 shows that with a calibrated predictor, the invertibility condition as required by BBSE (or RLLS) is exactly the same as the linear independence condition required for MLLS's consistency.

Having provided sufficient conditions, we consider a binary classification example to provide intuition for why we need calibration for consistency. In this example, we relate the estimation error to the miscalibration error, showing that calibration is not only sufficient but also necessary to achieve zero estimation error for a certain class of predictors.

**Example 1.** Consider a mixture of two Gaussians with $p_s(x|y = 0) := \mathcal{N}(\mu, 1)$ and $p_s(x|y = 1) := \mathcal{N}(-\mu, 1)$. We suppose that the source mixing coefficients are both $\frac{1}{2}$, while the target mixing

coefficients are $\alpha(\neq \frac{1}{2}), 1 - \alpha$. Assume a class of probabilistic threshold classifiers: $f(x) = [1 - c, c]$ for $x \geq 0$, otherwise $f(x) = [c, 1 - c]$ with $c \in [0, 1]$. Then the population error of MLLS is given by

$$4 \left| \frac{(1 - 2\alpha)(p_s(x \geq 0 | y = 0) - c)}{1 - 2c} \right|,$$

which is zero only if $c = p_s(x \geq 0 | y = 0)$ for a non-degenerate classifier.

The expression for estimation error arising from our example yields two key insights: (i) an uncalibrated thresholded classifier has an estimation error proportional to the true shift in label distribution i.e. $1 - 2\alpha$; (ii) the error is also proportional to the canonical calibration error which is $p_s(x \geq 0 | y = 0) - c$. While earlier in this section, we concluded that canonical calibration is sufficient for consistency, the above example provides some intuition for why it might also be necessary. In Appendix C, we show that marginal calibration [14, 10, 24], a less restricted definition is insufficient to achieve consistency.

### 4.4 MLLS with Confusion Matrix

So far, we have shown that MLLS with any calibrated predictor can be viewed as distribution matching in a latent space. Now we discuss a method to construct a predictor $f$ to perform MLLS given any $p(z|x)$, e.g., those induced by confusion matrix approaches. Recall, we already have the maximum log-likelihood objective. It just remains to construct a calibrated predictor $f$ from the confusion matrix.

This is straightforward when $p(z|x)$ is deterministic, i.e., $p(z|x) = \delta_{g(x)}$ for some function $g$: setting $f(x) = p_s(y|g(x))$ makes the objectives (3) and (4) to be the same. Recall that for the hard confusion matrix, the induced latent space is $p(z|x) = \delta_{\arg\max \widehat{f}(x)}$. So the corresponding predictor in MLLS is $f(x) = p_s(y|\widehat{y}_x)$, where $\widehat{y}_x = \arg\max \widehat{f}(x)$. Then we obtain the MLLS objective for the hard confusion matrix:

$$\max_{w \in \mathcal{W}} \mathbb{E}_t \left[ \log \sum\nolimits_{y=1}^{k} p_s(y|\widehat{y}_x) w_y \right]. \tag{6}$$

The confusion matrix $C_{\widehat{f}}$ and predictor $\widehat{f}$ directly give us $p_s(y|\widehat{y}_x)$. Given an input $x$, one can first get $\widehat{y}_x$ from $\widehat{f}$, then normalize the $\widehat{y}_x$-th row of $C_{\widehat{f}}$ as $p_s(y|\widehat{y}_x)$. We denote MLLS with hard confusion matrix calibration (6) by MLLS-CM.

When $p_s(z|x)$ is stochastic, we need to extend (4) to allow $f$ to be a random predictor: $f(x) = p_s(y|z)$ for $z \sim p(z|x)$[3]. To incorporate the randomness of $f$, one only needs to change the expectation in (4) to be over both $x$ and $f(x)$, then (4) becomes a rewrite of (3).

Proposition 2 indicates that constructing the confusion matrix is a calibration procedure. Thus, the predictor constructed with constructed using confusion matrix is calibrated and suitable for application with MLLS.

**Proposition 2** (Vaicenavicius et al. [24]). *For any function $g$, $f(x) = p_s(y|g(x))$ is a calibrated predictor.*

We can now summarize the relationship between BBSE and MLLS: A label shift estimator involves two design choices: (i) designing the latent space $p(z|x)$ (which is equivalent to designing a calibrated predictor); and (ii) performing distribution matching in the new space $\mathcal{Z}$. In BBSE, we design a calibrated predictor via the confusion matrix and then perform distribution matching by directly solving linear equations. In general, MLLS does not specify how to obtain a calibrated predictor, but specifies KL minimization as the distribution matching procedure. One can apply the confusion matrix approach to obtain a calibrated predictor and then plug it into MLLS, which is the BBSE analog under MLLS, and is a special case of MLLS.

## 5  Theoretical Analysis of MLLS

We now analyze the performance of MLLS estimator. Even when $w^*$ is the unique optimizer of (4) for some calibrated predictor $f$, assuming convex optimization can be done perfectly, there are still two sources of error preventing us from exactly computing $w^*$ in practice. First, we are optimizing a sample-based approximation (5) to the objective in expectation (4). We call this source of error

*finite-sample error.* Second, the predictor $f$ we use may not be perfectly calibrated on the source distribution as we only have access to samples from source data distribution $p_s(x, y)$. We call this source of error *miscalibration error*. We will first analyze how these two sources of errors affect the estimate of $w^*$ separately and then give a general error bound that incorporates both. All proofs are relegated to Appendix E.

Before presenting our analysis, we introduce some notation and regularity assumptions. For any predictor $f : \mathcal{X} \mapsto \Delta^{k-1}$, we define $w_f$ and $\widehat{w}_f$ as in (4) and (5). If $f$ satisfies the conditions in Theorem 2 (calibration and linear independence) then we have that $w_f = w^*$. Our goal is to bound $\|\widehat{w}_f - w^*\|$ for a given (possibly miscalibrated) predictor $f$. We now introduce a regularity condition:

**Condition 1** (Regularity condition for a predictor $f$). *For any $x$ within the support of $p_t(x)$, i.e.* $p_t(x) > 0$, *we have both $f(x)^T w_f \geq \tau$, $f(x)^T w^* \geq \tau$ for some universal constant $\tau > 0$.*

Condition 1 is mild if $f$ is calibrated since in this case $w_f = w^*$ is the maximizer of $\mathbb{E}_t \left[ \log f(x)^T w \right]$, and the condition is satisfied if the expectation is finite. Since $f(x)^T w^*$ and $f(x)^T w_f$ are upper-bounded (they are the inner products of two vectors which sum to 1), they also must be lower-bounded away from 0 with arbitrarily high probability without any assumptions. For miscalibrated $f$, a similar justification holds for assumption that $f(x)^T w_f$ is lower bounded. Turning our attention to the assumption that $f(x)^T w^*$ is lower bounded, we note that it is sufficient if $f$ is close (pointwise) to some calibrated predictor. This in turn is a reasonable assumption on the actual predictor we use for MLLS in practice as it is post-hoc calibrated on source data samples.

Define $\sigma_{f,w}$ to be the minimum eigenvalue of the Hessian $-\nabla_w^2 \mathcal{L}(w, f)$. To state our results compactly we use standard stochastic order notation (see, for instance, [26]). We first bound the estimation error introduced by only having finite samples from the target distribution in Lemma 3. Next, we bound the estimation error introduced by having a miscalibrated $f$ in Lemma 4.

**Lemma 3.** *For any predictor $f$ that satisfies Condition 1, we have $\|w_f - \widehat{w}_f\| \leq \sigma_{f,w_f}^{-1} \mathcal{O}_p \left( m^{-1/2} \right)$.*

**Lemma 4.** *For any predictor $f$ and any calibrated predictor $f_c$ that satisfies Condition 1, we have* $\|w_f - w^*\| \leq \sigma_{f,w^*}^{-1} \cdot C \cdot \mathbb{E}_t \left[ \|f - f_c\| \right]$, *for some constant $C$.*

*If we set $f_c(x) = p_s(y|f(x))$, which is a calibrated predictor (Proposition 2), we can bound the error in terms of the calibration error of $f$ on the source data* [4]: $\|w_f - w^*\| \leq \sigma_{f,w^*}^{-1} \cdot C \cdot \mathcal{E}(f)$.

Note that since $p_s(y) > 0$ for all $y$, we can upper-bound the error in Lemma 4 with calibration error on the source data. We combine the two sources of error to bound the estimation error $\|\widehat{w}_f - w^*\|$:

**Theorem 3.** *For any predictor $f$ that satisfies Condition 1, we have*

$$\|\widehat{w}_f - w^*\| \leq \sigma_{f,w_f}^{-1} \mathcal{O}_p \left( m^{-1/2} \right) + C \cdot \sigma_{f,w^*}^{-1} \mathcal{E}(f) . \tag{7}$$

The estimation error of MLLS can be decomposed into (i) finite-sample error, which decays at a rate of $m^{-1/2}$; and (ii) the calibration error of the predictor that we use. The proof is a direct combination of Lemma 3 and Lemma 4 applied to the same $f$ with the following error decomposition:

$$\|\widehat{w}_f - w^*\| \leq \underbrace{\|w_f - \widehat{w}_f\|}_{\text{finite-sample}} + \underbrace{\|w_f - w^*\|}_{\text{miscalibration}} .$$

Theorem 3 shows that the estimation error depends inversely on the minimum eigenvalue of the Hessian at two different points $w_f$ and $w^*$. One can unify these two eigenvalues as a single quantity $\sigma_f$, the minimum eigenvalue $\mathbb{E}_t \left[ f(x)f(x)^T \right]$. We formalize this observation in Appendix E.

If we use the *post-hoc calibration* procedure (as discussed in Section 2 and A) to calibrate a blackbox predictor $\widehat{f}$, we can obtain a bound on the calibration error of $f$. In more detail, suppose that the class $\mathcal{G}$ used for calibration satisfies standard regularity conditions (injectivity, Lipschitz-continuity, twice differentiability, non-singular Hessian). We have the following lemma:

**Lemma 5.** *Let $f = g \circ \widehat{f}$ be the predictor after post-hoc calibration with squared loss $l$ and $g$ belongs to a function class $\mathcal{G}$ that satisfies the standard regularity conditions, we have*

$$\mathcal{E}(f) \leq \min_{g \in \mathcal{G}} \mathcal{E}(g \circ \widehat{f}) + \mathcal{O}_p \left( n^{-1/2} \right) . \tag{8}$$

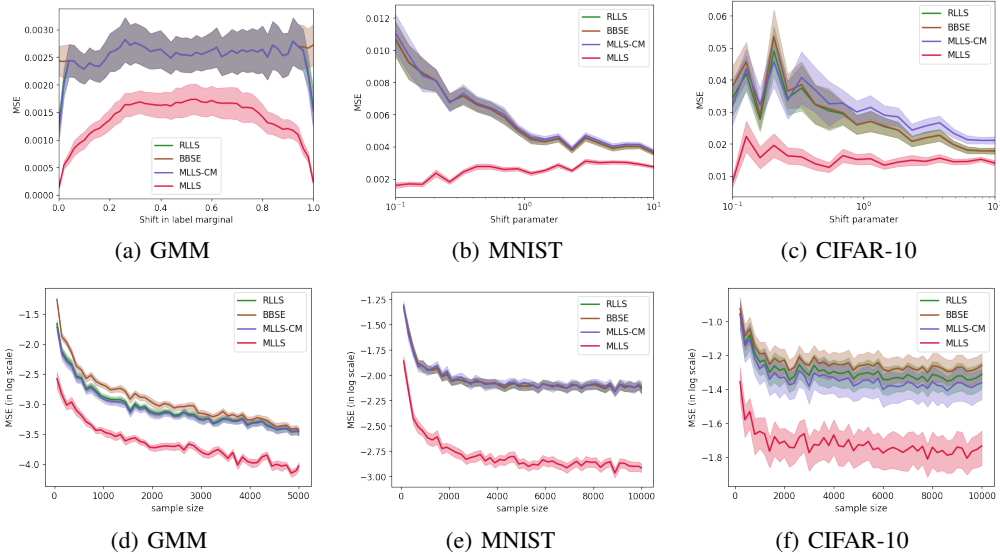

(a) GMM  (b) MNIST  (c) CIFAR-10

(d) GMM  (e) MNIST  (f) CIFAR-10

Figure 1: (**top**) MSE vs the degree of shift; For GMM, we control the shift in the label marginal for class 1 with a fixed target sample size of 1000. For multiclass problems—-MNIST and CIFAR-10, we control the Dirichlet shift parameter with a fixed sample size of 5000. (**bottom**) MSE (in log scale) vs target sample size; For GMM, we fix the label marginal for class 1 at 0.01 whereas for multiclass problems, MNIST and CIFAR-10, we fix the Dirichlet parameter to 0.1. In all plots, MLLS dominates other methods. All confusion matrix approaches perform similarly, indicating that the advantage of MLLS comes from the choice of calibration but not the way of performing distribution matching.

This result is similar to Theorem 4.1 [14]. For a model class $\mathcal{G}$ that is rich enough to contain a function $g \in \mathcal{G}$ that achieves zero calibration error, i.e., $\min_{g \in \mathcal{G}} \mathcal{E}(g \circ \widehat{f}) = 0$, then we obtain an estimation error bound for MLLS of $\sigma_f^{-1} \cdot \mathcal{O}_p\left(m^{-1/2} + n^{-1/2}\right)$. This bound is similar to rate of RLLS and BBSE, where instead of $\sigma_f$ they have minimum eigenvalue of the confusion matrix.

The estimation error bound explains the efficiency of MLLS. Informally, the error of MLLS depends inversely on the minimum eigenvalue of the Hessian of the likelihood $\sigma_f$. When we apply coarse calibration via the confusion matrix (in MLLS-CM), we only decrease the value of $\sigma_f$. Coarse calibration throws away information [13] and thus results in greater estimation error for MLLS. In Section 6, we emprically show that MLLS-CM's performance is similar to that of BBSE. Moreover, on a synthetic Gaussian mixture model, we show that the minimum eigenvalue of the Hessian obtained using confusion matrix calibration is smaller than the minimum eigenvalue obtained with more granular calibration. Our analysis and observations together suggest MLLS's superior performance than BBSE (or RLLS) is due to the granular calibration but not due to the difference in the optimization objective.

Finally, we want to highlight one minor point regarding applicability of our result. If $f$ is calibrated, Theorem 3, together with Proposition 3 (in Appendix E), implies that MLLS is consistent if $\mathbb{E}_t \left[ f(x)f(x)^T \right]$ is invertible. Compared to the consistency condition in Theorem 1 that $\mathbb{E}_s \left[ f(x)f(x)^T \right]$ is invertible (together with Proposition 1), these two conditions are the same if the likelihood ratio $p_t(f(x))/p_s(f(x))$ is lower-bounded. This is true if all entries in $w^*$ are non-zero. Even if $w^*$ contains non-zero entries, the two conditions are still the same if there exists some $w_y^* > 0$ such that $p(f(x)|y)$ covers the full support of $p_s(f(x))$. In general however, the invertibility of $\mathbb{E}_t \left[ f(x)f(x)^T \right]$ is a stronger requirement than the invertibility of $\mathbb{E}_s \left[ f(x)f(x)^T \right]$. We leave further investigation of this gap for future work.

## 6 Experiments

We experimentally illustrate the performance of MLLS on synthetic data, MNIST [15], and CIFAR10 [12]. Following Lipton et al. [16], we experiment with *Dirichlet shift* simulations. On each run, we sample a target label distribution $p_t(y)$ from a Dirichlet with concentration parameter $\alpha$. We

then generate each target example by first sampling a label $y \sim p_t(y)$ and then sampling (with replacement) an example conditioned on that label . Note that smaller values of alpha correspond to more severe shift. In our experiments, the source label distribution is uniform.

First, we consider a mixture of two Gaussians (as in Example in Section 4.3) with $\mu = 1$. With CIFAR10 and MNIST, we split the full training set into two subsets: train and valid, and use the provided test set as is. Then according to the label distribution, we randomly sample with replacement train, valid, and test set from each of their respective pool to form the source and target set. To learn the black box predictor on real datasets, we use the same architecture as Lipton et al. [16] for MNIST, and for CIFAR10 we use ResNet-18 [11] as in Azizzadenesheli et al. [2][5]. For simulated data, we use the true $p_s(y|x)$ as our predictor function. For each experiment, we sample 100 datasets for each shift parameter and evaluate the empirical MSE and variance of the estimated weights.

We consider three sets of experiments: (1) MSE vs degree of target shift; (2) MSE vs target sample sizes; and (3) MSE vs calibrated predictors on the source distribution. We refer to MLLS-CM as MLLS with hard confusion matrix calibration as in (6). In our experiments, we compare MLLS estimator with BBSE, RLLS, and MLLS-CM. For RLLS and BBSE, we use the publicly available code [6]. To post-hoc calibration, we use BCTS [1] on the held-out validation set. Using the same validation set, we calculate the confusion matrix for BBSE, RLLS, and MLLS-CM.

We examine the performance of various estimators across all three datasets for various target dataset sizes and shift magnitudes (Figure 1). Across all shifts, MLLS (with BCTS-calibrated classifiers) *uniformly dominates* BBSE, RLLS, and MLLS-CM in terms of MSE (Figure 1). Observe for severe shifts, MLLS is comparatively dominant. As the available target data increased, all methods improve rapidly, with MLLS outperforming all other methods by a significant margin. Moreover, MLLS's advantages grow more pronounced under extreme shifts. Notice MLLS-CM is roughly equivalent to BBSE across all settings of dataset, target size, and shift magnitude. This concludes MLLS's superior performance is not because of differences in loss function used for distribution matching but due to differences in the granularity of the predictions, caused by crude confusion matrix aggregation.

Note that given a predictor $f_1$, we can partition our input space and produce another predictor $f_2$ that, for any datapoint gives the expected output of $f_1$ on points belonging to that partition. If $f_1$ is calibrated, then $f_2$ will also be calibrated [24]. On synthetic data, we vary the granularity of calibration (for MLLS) by aggregating $p_s(y|x)$ over a variable number of equal-sized bins. With more bins, less information is lost due to calibration. Consequently, the minimum eigenvalue of the Hessian increases and the MSE decreases, supporting our theoretical bounds (Figure 2). We also verify that the confusion matrix calibration performs poorly (Figure 2). For MLLS-CM, the minimum eigenvalue of the Hessian is 0.195, significantly smaller than for the binned predictor for #bin $\geq 4$. Thus, the poor performance of MLLS-CM is predicted by its looser upper bound per our analysis. Note that these experiments presume access to the true predictor $p_s(y|x)$ and thus the

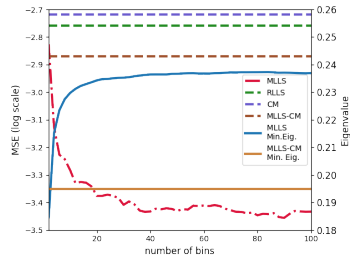

Figure 2: MSE (left-axis) with variation of minimum eigenvalue of the Hessian (right-axis) vs number of bins used for aggregation. With increase in number of bins, MSE decrease and the minimum eigenvalue increases.

MSE strictly improves with the number of bins. In practice, with a fixed source dataset size, increasing the number of bins could lead to overfitting, worsening our calibration.

# 7 Conclusion

This paper provides a unified framework relating techniques that use off-the-shelf predictors for label shift estimation. We argue that these methods all employ calibration, either explicitly or implicitly, differing only in the choice of calibration method and their optimization objective. Moreover, with our analysis we show that the choice of calibration method (and not the optimization objective for distribution matching) accounts for the advantage of MLLS with BCTS calibration over BBSE. In future work, we hope to operationalize these insights to provide guidance for a calibration scheme to improve label shift estimation.

## Broader Impact

This paper investigates the (statistical) consistency and efficiency of two existing methods for estimating target domain label distributions. While this could potentially guide practitioners to improve detection, estimation, and classification in applications where the label shift assumption holds, we do not believe that it will fundamentally impact how machine learning is used in a way that could conceivably be socially salient. While we take the potential impact of machine learning on society seriously, we believe that this work, which addresses a foundational theoretical problem, does not present a significant societal concern.

## Acknowledgments

We thank Zico Kolter and David Childers for their helpful feedback. This material is based on research sponsored by Air Force Research Laboratory (AFRL) under agreement number FA8750-19-1-1000. The U.S. Government is authorized to reproduce and distribute reprints for Government purposes notwithstanding any copyright notation therein. The views and conclusions contained herein are those of the authors and should not be interpreted as necessarily representing the official policies or endorsements, either expressed or implied, of Air Force Laboratory, DARPA or the U.S. Government. SB acknowledges funding from the NSF grants DMS-1713003 and CIF-1763734. ZL acknowledges Amazon AI, Salesforce Research, Facebook, UPMC, Abridge, and the Center for Machine Learning and Health for their generous support of ACMI Lab's research on machine learning under distribution shift.

## Footnotes

[1] Motivated by the strong empirical results in Alexandari et al. [1], we use BCTS in our experiments as a surrogate to canonical calibration.

[2]For simplicity we will use $z = x$ to denote that $p(z|x) = \delta_x$.

[3]Here, by a random predictor we mean that the predictor outputs a random vector from $\Delta^{k-1}$, not $\mathcal{Y}$.

[4]We present two upper bounds because the second is more interpretable while the first is tighter.

[5]We used open source implementation of ResNet-18 https://github.com/kuangliu/pytorch-cifar.

[6]BBSE: https://github.com/zackchase/label_shift, RLLS: https://github.com/Angela0428/labelshift

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
