[Supplementary Material]

# A MLLS Algorithm

---

**Algorithm 1** Maximum Likelihood Label Shift estimation

---

**input** : Labeled validation samples from source and unlabeled test samples from target. Trained blackbox model $\hat{f}$, model class $\mathcal{G}$ and loss function $l$ for calibration (for instance, MSE or negative log-likelihood).

1: On validation data minimize the loss $l$ over class $\mathcal{G}$ to obtain $f = g \circ \hat{f}$.

2: Solve the optimization problem (5) using $f$ to get $\widehat{w}$.

**output** : MLLS estimate $\widehat{w}$

---

**Step 1. description.** Let the model class used for *post-hoc calibration* be represented by $\mathcal{G}$. Given a validation dataset $\{(x_{v1}, y_{v1}), \ldots, (x_{vn}, y_{vn})\}$ sampled from the source distribution $P_s$ we compute, $\{(\widehat{f}(x_{v1}), y_{v1}), (\widehat{f}(x_{v2}), y_{v2}), \ldots, (\widehat{f}(x_{vn}), y_{vn})\}$, applying our classifier $\widehat{f}$ to the data. Using this we estimate a function,

$$\widehat{g} = \arg\min_{g \in \mathcal{G}} \sum_{i=1}^{n} \ell(g \circ \widehat{f}(x_{vi}), y_{vi}), \tag{9}$$

where the loss function $\ell$ can be the negative log-likelihood or squared error. Experimentally we observe same performance with both the loss functions. Subsequently, we can apply the calibrated predictor $\widehat{g} \circ \widehat{f}$.

Our experiments follow Alexandari et al. [1], who leverage BCTS [7] to calibrate their models. BCTS extends temperature scaling [10] by incorporating per-class bias terms. Formally, a function $g : \Delta^{k-1} \mapsto \Delta^{k-1}$ in the BCTS class $\mathcal{G}$, is given by

$$g_j(x) = \frac{\exp\left[\log(x_j)/T + b_j\right]}{\sum_i \exp\left[\log(x_i)/T + b_i\right]} \quad \forall j \in \mathcal{Y}$$

where $\{T, b_1, \ldots, b_{|\mathcal{Y}|}\}$ are the $|\mathcal{Y}| + 1$ parameters to be learned.

# B Prior Work on Label Shift Estimation

Dataset shifts are predominantly studied under two scenarios: covariate shift and label shift [22]. Schölkopf et al. [19] articulates connections between label shift and covariate shift with anti-causal and causal models respectively. Covariate shift is well explored in past [28, 27, 7, 6, 9].

Approaches for estimating label shift (or prior shift) can be categorized into three classes:

1. Methods that leverage Mixture Proportion Estimation (MPE) [4, 17] techniques to estimate the target label distribution. MPE estimate in general (e.g. Blanchard et al. [4]) needs explicit calculations of $p_s(x|y)(= p_t(x|y))$ which is infeasible for high dimensional data. More recent methods for MPE estimation, i.e. Ramaswamy et al. [17], uses Kernel embeddings, which like many kernel methods, require the inversion of an $n \times n$ Gram matrix. The $\mathcal{O}(n^3)$ complexity makes them infeasible for large datasets, practically used in deep learning these days;

2. Methods that directly operate in RKHS for distribution matching [28, 8]. Zhang et al. [28] extend the kernel mean matching approach due to Gretton et al. [9] to the label shift problem. Instead of minimizing maximum mean discrepancy, Du Plessis and Sugiyama [8] explored minimizing PE divergence between the kernel embeddings to estimate the target label distribution. Again, both the methods involve inversion of an $n \times n$ kernel matrix, rendering them infeasible for large datasets; and

3. Methods that work in low dimensional setting [16, 2, 18] by directly estimating $p_t(y)/p_s(y)$ to avoid the curse of dimensionality. These methods leverage an off-the-shelf predictor to estimate the label shift ratio.

In this paper, we primarily focus on unifying methods that fall into the third category.

## C  Marginal calibration is insufficient to achieve consistency

In this section, we will illustrate insufficiency of *marginal calibration* to achieve consistency. For completeness, we first define margin calibration:

**Definition 3** (Marginal calibration). *A prediction model $f : \mathcal{X} \mapsto \Delta^{k-1}$ is marginally calibrated on the source domain if for all $x \in \mathcal{X}$ and $j \in \mathcal{Y}$,*

$$\mathrm{P}_s(y = j | f_j(x)) = f_j(x) \,.$$

Intuitively, this definition captures per-label calibration of the classifier which is strictly less restrictive than requiring canonical calibration. In the example, we construct a classifier on discrete $\mathcal{X}$ which is marginally calibrated, but not canonically calibrated. With the constructed example, we show that the population objective (4) yields inconsistent estimates.

**Example**. Assume $\mathcal{X} = \{x_1, x_2, x_3, x_4, x_5, x_6\}$ and $\mathcal{Y} = \{1, 2, 3\}$. Suppose the predictor $f(x)$ and $P_s(y|f(x))$ are given as,

| $f(x)$ | y=1 | y=2 | y=3 |
|---|---|---|---|
| $x_1$ | 0.1 | 0.2 | 0.7 |
| $x_2$ | 0.1 | 0.7 | 0.2 |
| $x_3$ | 0.2 | 0.1 | 0.7 |
| $x_4$ | 0.2 | 0.7 | 0.1 |
| $x_5$ | 0.7 | 0.1 | 0.2 |
| $x_6$ | 0.7 | 0.2 | 0.1 |

| $P_s(y|f(x))$ | y=1 | y=2 | y=3 |
|---|---|---|---|
| $x_1$ | 0.2 | 0.1 | 0.7 |
| $x_2$ | 0.0 | 0.8 | 0.2 |
| $x_3$ | 0.1 | 0.2 | 0.7 |
| $x_4$ | 0.3 | 0.6 | 0.1 |
| $x_5$ | 0.8 | 0.0 | 0.2 |
| $x_6$ | 0.6 | 0.3 | 0.1 |

Clearly, the prediction $f(x)$ is marginally calibrated. We have one more degree to freedom to choose, which is the source marginal distribution on $\mathcal{X}$. For simplicity let's assume $p_s(x_i) = 1/6$ for all $i = \{1, \ldots, 6\}$. Thus, we have $p_s(y = j) = 1/3$ for all $j = \{1, 2, 3\}$. Note, with our assumption of the source marginal on x, we get $P_t(x_i|y = j) = P_s(x_i|y = j) = P_s(y = j|f(x_i))/2$. This follows as $x \mapsto f(x)$ is an one-to-one mapping.

Now, assume a shift i.e. prior on $\mathcal{Y}$ for the target distribution of the form $[\alpha, \beta, 1 - \alpha - \beta]$. With the label shift assumption, we get

$$\forall i \qquad p_t(x_i) = \frac{1}{2} \left( \alpha P_s(y = 1|f(x_i)) + \beta P_s(y = 2|f(x_i)) + (1 - \beta - \alpha)P_s(y = 3|f(x_i)) \right) \,.$$

Assume the importance weight vector as $w$. Clearly, we have $w_1 + w_2 + w_3 = 3$. Re-writing the population MLLS objective (4), we get the maximisation problem as

$$\arg\max_{w} \sum_{i=1}^{6} p_t(x_i) \log(f(x_i)^T w) \,. \tag{10}$$

Differentiating (10) with respect to $w_1$ and $w_2$, we get two high order equations, solving which give us the MLLS estimate $w_f$. To show inconsistency, it is enough to consider one instantiation of $\alpha$ and $\beta$ such that $|3\alpha - w_1| + |3\beta - w_2| + |w_1 + w_2 - 3\alpha - 3\beta| \neq 0$. Assuming $\alpha = 0.8$ and $\beta = 0.1$ and solving (10) using numerical methods, we get $w_f = [2.505893, 0.240644, 0.253463]$. As $w = [2.4, 0.3, 0.3]$, we have $w_f \neq w$ concluding the proof.

## D  Proofs from Section 4

**Lemma 1** (Identifiability). *If the set of distributions $\{p(z|y) : y = 1, ..., k\}$ are linearly independent, then for any $w$ that satisfies (2), we must have $w = w^*$. This condition is also necessary in general: if the linear independence does not hold then there exists a problem instance where we have $w, w^* \in \mathcal{W}$ satisfying (2) while $w \neq w^*$.*

*Proof.* First we prove sufficiency. If there exists $w \neq w^*$ such that (2) holds, then we have $\sum_{y=1}^{k} p_s(z, y)(w_y - w_y^*) = 0$ for all $z \in \mathcal{Z}$. As $w - w^*$ is not the zero vector, $\{p_s(z, y), y = 1, ..., k\}$ are linearly dependent. Since $p_s(z, y) = p_s(y)p(z|y)$ and $p_s(y) > 0$ for all $y$ (by assumption), we

also have that $\{p(z|y), y = 1, ..., k\}$ are linearly dependent. By contradiction, we show that the linear independence is necessary.

To show necessity, assume $w_y^* = \frac{1}{k p_s(y)}$ for $y = 1, ..., k$. We know that $w^*$ satisfies (2) by definition. If linear independence does not hold, then there exists a vector $v \in \mathbb{R}^k$ such that $v \neq 0$ and $\sum_{y=1}^{k} p_s(z, y) v_y = 0$ for all $z \in \mathcal{Z}$. Since the $w^*$ we construct is not on the boundary of $\mathcal{W}$, we can scale $v$ such that $w^* + \alpha v \in \mathcal{W}$ where $\alpha \geq 0$ and $v \neq 0$. Therefore, setting $w = w^* + \alpha v$ gives another solution for (2), which concludes the proof. □

**Lemma 2.** *If $f$ is calibrated, then the two objectives* (3) *and* (4) *are identical when $\mathcal{Z}$ is chosen as $\Delta^{k-1}$ and $p(z|x)$ is defined to be $\delta_{f(x)}$.*

*Proof.* The proof follows a sequence of straightforward manipulations. In more detail,

$$
\begin{aligned}
\mathbb{E}_t \left[ \log f(x)^T w \right] &= \int p_t(x) \log[f(x)^T w] dx \\
&= \int \int p_t(x) p(z|x) \log[f(x)^T w] dx dz \\
&= \int \int p_t(x) p(z|x) \mathbb{1}\{f(x) = z\} \log[f(x)^T w] dx dz \\
&= \int \int p_t(x) p(z|x) \log[z^T w] dx dz \\
&= \int p_t(z) \log[z^T w] dz \\
&= \int p_t(z) \log \left[ \sum_{y=1}^{k} p_s(y|z) w \right] dz \,,
\end{aligned}
$$

where the final step uses the fact that $f$ is calibrated.

□

**Theorem 1** (Population consistency of MLLS). *If a predictor $f : \mathcal{X} \mapsto \Delta^{k-1}$ is calibrated and the distributions $\{p(f(x)|y) : y = 1, \ldots, k\}$ are strictly linearly independent, then $w^*$ is the unique maximizer of the MLLS objective* (4).

*Proof.* According to Lemma 2 we know that maximizing (4) is the same as maximizing (3) with $p(z|x) = \delta_{f(x)}$, thus also the same as minimizing the KL divergence between $p_t(z)$ and $p_w(z)$. Since $p_t(z) \equiv p_{w^*}(z)$ we know that $w^*$ is a minimizer of the KL divergence such that the KL divergence is 0. We also have that $\mathrm{KL}(p_t(z), p_w(z)) = 0$ if and only if $p_t(z) \equiv p_w(z)$, so all maximizers of (4) should satisfy (2). According to Lemma 1, if the strict linear independence holds, then $w^*$ is the unique solution of (2). Thus $w^*$ is the unique maximizer of (4).

□

**Proposition 1.** *For a calibrated predictor $f$, the following statements are equivalent:*

    *(1) $\{p(f(x)|y) : y = 1, \ldots, k\}$ are strictly linearly independent.*
    *(2) $\mathbb{E}_s \left[ f(x) f(x)^T \right]$ is invertible.*
    *(3) The soft confusion matrix of $f$ is invertible.*

*Proof.* We first show the equivalence of (1) and (2). If $f$ is calibrated, we have $p_s(f(x)) f_y(x) = p_s(y) p(f(x)|y)$ for any $x, y$. Then for any vector $v \in \mathbb{R}^k$ we have

$$
\sum_{y=1}^{k} v_y p(f(x)|y) = \sum_{y=1}^{k} \frac{v_y}{p_s(y)} p_s(y) p(f(x)|y) = \sum_{y=1}^{k} \frac{v_y}{p_s(y)} p_s(f(x)) f_y(x) = p_s(f(x)) \sum_{y=1}^{k} \frac{v_y}{p_s(y)} f_y(x) \,.
$$
$$(11)$$

On the other hand, we can have

$$\mathbb{E}_s\left[f(x)f(x)^T\right] = \int f(x)f(x)^T p_s(f(x))d(f(x)).\tag{12}$$

If $\{p(f(x)|y) : y = 1,\ldots,k\}$ are linearly dependent, then there exist $v \neq 0$ such that (11) is zero for any $x$. Consequently, there exists a non-zero vector $u$ with $u_y = v_y/p_s(y)$ such that $u^T f(x) = 0$ for any $x$ satisfying $p_s(f(x)) > 0$, which means $u^T\mathbb{E}_s\left[f(x)f(x)^T\right]u = 0$ and thus $\mathbb{E}_s\left[f(x)f(x)^T\right]$ is not invertible. On the other hand, if $\mathbb{E}_s\left[f(x)f(x)^T\right]$ is non-invertible, then there exist some $u \neq 0$ such that $u^T\mathbb{E}_s\left[f(x)f(x)^T\right]u = 0$. Further as $u^T\mathbb{E}_s\left[f(x)f(x)^T\right]u = \int u^T f(x)f(x)^T u\, p_s(x)dx = \int\left|f(x)^T u\right|p_s(x)dx$. As a result, the vector $v$ with $v_y = p_s(y)u_y$ satisfies that (11) is zero for any $x$, which means $\{p(f(x)|y) : y = 1,\ldots,k\}$ are not strictly linearly independent.

Let $C$ be the soft confusion matrix of $f$, then

$$C_{ij} = p_s(\widehat{y} = i, y = j) = \int d(f(x))\, f_i(x)p(f(x)|y = j)p_s(y = j)$$

$$= \int f_i(x)f_j(x)p_s(f(x))d(f(x)).$$

Therefore, we have $C = \mathbb{E}_s\left[f(x)f(x)^T\right]$, which means (2) and (3) are equivalent.

$\square$

We introduce some notation before proving consistency. Let $\mathscr{P} = \{\langle f, w\rangle|w \in \mathcal{W}\}$ be the class of densities[8] for a given calibrated predictor $f$. Suppose $\widehat{p}_n, p_0 \in \mathscr{P}$ are densities corresponding to MLE estimate and true weights, respectively. We use $h(p_1, p_2)$ to denote the Hellinger distance and $\mathrm{TV}(p_1, p_2)$ to denote the total variation distance between two densities $p_1, p_2$. $H_r(\delta, \mathscr{P}, P)$ denotes $\delta$-entropy for class $\mathscr{P}$ with respect to metric $L_r(P)$. Similarly, $H_{r,B}(\delta, \mathscr{P}, P)$ denotes the corresponding bracketing entropy. Moreover, $P_n$ denotes the empirical random distribution that puts uniform mass on observed samples $x_1, x_2, \ldots x_n$. Before proving consistency we need to re-state two results:

**Lemma 6** (Lemma 2.1 [25]). *If $P$ is a probability measure, for all $1 \leq r < \infty$, we have*

$$H_{r,B}(\delta, \mathscr{G}, P) \leq H_\infty(\delta/2, \mathscr{G})\qquad\text{for all } \delta > 0.$$

**Lemma 7** (Corollary 2.7.10 [26]). *Let $\mathcal{F}$ be the class of convex functions $f : C \mapsto [0, 1]$ defined on a compact, convex set $C \subset \mathbb{R}^d$ such that $|f(x) - f(y)| \leq L\|x - y\|$ for every x,y. Then*

$$H_\infty(\delta, \mathcal{F}) \leq K\left(\frac{L}{\delta}\right)^{d/2},$$

*for a constant K that depends on the dimension d and C.*

We can now present our proof of consistency, which is based on Theorem 4.6 from van de Geer [25]:

**Lemma 8** (Theorem 4.6 [25]). *Let $\mathscr{P}$ be convex and define class $\mathscr{G} = \left\{\frac{2p}{p+p_0}|p \in \mathscr{P}\right\}$. If*

$$\frac{1}{n}H_1(\delta, \mathscr{G}, P_n) \to_P 0,\tag{13}$$

*then $h(\widehat{p}_n, p_0) \to 0$ almost surely.*

**Theorem 2** (Consistency of MLLS). *If $f$ satisfies the conditions in Theorem 1, then $\widehat{w}_f$ in (5) converges to $w^*$ almost surely.*

*Proof.* Assume the maximizer of (5) is $\widehat{w}_f$ and $p_0 = \langle f, w^*\rangle$. Define class $\mathscr{G} = \left\{\frac{2p}{p+p_0}|p \in \mathscr{P}\right\}$. To prove consistency, we first bound the bracketing entropy for class $\mathscr{G}$ using Lemma 6 and Lemma 7.

Clearly $\mathscr{P}$ is linear in parameters and hence, convex. Gradient of function $g \in \mathscr{G}$ is given by $\frac{2p_0}{(p+p_0)^2}$ which in turn is bounded by $\frac{2}{p_0}$. Under assumptions of Condition 1, the functions in $\mathscr{G}$ are Lipschitz with constant $2/\tau$. We can bound the bracketing entropy $H_{2,B}(\delta, \mathscr{G}, P)$ using Lemma 7 and Lemma 6 as

$$H_{2,B}(\delta, \mathscr{G}, P) \leq H_\infty(\delta, \mathscr{G}) \leq K_1 \left(\frac{1}{\delta\tau}\right)^{k/2},$$

for some constant $K_1$ that depends on $k$.

On the other hand, for cases where $p_0$ can be arbitrarily close to zero, i.e., Condition 1 doesn't hold true, we define $\tau(\delta)$ and $\mathscr{G}_\tau$ as

$$\tau(\delta) = \sup\left\{\tau \geq 0 \mid \int_{p_0 \leq \tau} p_0 dx \leq \delta^2\right\}, \tag{14}$$

$$\mathscr{G}_\tau = \left\{\frac{2p}{p+p_0}\mathbb{1}\{p_0 \geq \tau\} \mid p \in \mathscr{P}\right\}.$$

Using triangle inequality, for any $g_1, g_2 \in \mathscr{G}$, we have

$$\int \|g_1 - g_2\|^2 dx \leq \int \|g_1 - g_2\|^2 \mathbb{1}\{p_0 \leq \tau\} dx + \int \|g_1 - g_2\|^2 \mathbb{1}\{p_0 \geq \tau\} dx$$

$$\leq 2 \int \mathbb{1}\{p_0 \leq \tau\} dx + \int \|g_1 - g_2\|^2 \mathbb{1}\{p_0 \geq \tau\} dx. \tag{15}$$

Assume $\tau(\delta)$ such that (14) is satisfied. Using (15), we have

$$H_{2,B}(\delta, \mathscr{G}, P) \leq H_{2,B}(\sqrt{3}\delta, \mathscr{G}_{\tau(\delta)}, P).$$

Thus, for the cases where $p_0$ can be arbitrarily close to zero, instead of bounding $H_{2,B}(\delta, \mathscr{G}, P)$, we we bound $H_B(\delta, \mathscr{G}_{\tau(\delta)}, P)$. For any $\delta > 0$, there is a compact subset $K_\delta \in \mathcal{X}$, such that $p_s(X \setminus K_\delta) < \delta$. Using arguments similar to above, function $g \in \mathscr{G}_{\tau(\delta)}$ is Lipschitz with constant $2/\tau(\delta) > 0$. Again using Lemma 7 and Lemma 6, we conclude

$$H_{2,B}(2\delta, \mathscr{G}_{\tau(\delta)}, P) \leq H_\infty(\delta, \mathscr{G}_{\tau(\delta)}) \leq K_2 \left(\frac{1}{\delta\tau(\delta)}\right)^k,$$

for some constant $K_2$ that depends on $k$. Finally, we use Lemma 8 to conclude $h(\widehat{p}_n, p_0) \to_{\text{a.s.}} 0$. Further, as $\text{TV}(\widehat{p}_n, p_0) \leq h(\widehat{p}_n, p_0)$, we have $h(\widehat{p}_n, p_0) \to_{\text{a.s.}} 0$ implies $\text{TV}(\widehat{p}_n, p_0) \to_{\text{a.s.}} 0$. Further

$$\|\widehat{w}_f - w^*\|^2 \leq \frac{1}{\lambda_{\min}} \int \left|f(x)^T(\widehat{w}_f - w^*)\right|^2 p_s(x)dx$$

$$\leq \frac{\sup_x \left\{\left|f(x)^T(\widehat{w}_f - w^*)\right|\right\}}{\lambda_{\min}} \underbrace{\int \left|f(x)^T(\widehat{w}_f - w^*)\right| p_s(x)dx}_{\text{TV}(\widehat{p}_n, p_0)}, \tag{16}$$

where $\lambda_{\min}$ is the minimum eigenvalue of covariance matrix $\left[\int f(x)f(x)^T p_s(x)dx\right]$. Note using Proposition 1, we have $\lambda_{\min} > 0$. Thus, we conclude $\|\widehat{w}_f - w^*\| \to_{\text{a.s.}} 0$. $\qquad\square$

**Example 1.** Consider a mixture of two Gaussians with $p_s(x|y = 0) := \mathcal{N}(\mu, 1)$ and $p_s(x|y = 1) := \mathcal{N}(-\mu, 1)$. We suppose that the source mixing coefficients are both $\frac{1}{2}$, while the target mixing coefficients are $\alpha(\neq \frac{1}{2}), 1 - \alpha$. Assume a class of probabilistic threshold classifiers: $f(x) = [1 - c, c]$ for $x \geq 0$, otherwise $f(x) = [c, 1 - c]$ with $c \in [0, 1]$.

Then the population error of MLLS is given by

$$4\left|\frac{(1 - 2\alpha)(p_s(x \geq 0|y = 0) - c)}{1 - 2c}\right|,$$

which is zero only if $c = p_s(x \geq 0|y = 0)$ for a non-degenerate classifier.

*Proof.* The intuition behind the construction is, for such an Example, we can get a closed form solution for the population MLLS and hence allows a careful analysis of the estimation error. The classifier $f(x)$ predicts class 0 with probability $c$ and class 1 with probability $1 - c$ for $x \geq 0$, and vice-versa for $x < 0$. Using such a classifier, the weight estimator is given by:

$$\widehat{w} = \underset{w}{\arg\min}\, \mathbb{E}\left[\log\langle f(x), w\rangle\right]$$

$$\overset{(i)}{=} \underset{w_0}{\arg\min}\left[\int_{-\infty}^{0} \log((1-c)w_0 + c(2-w_0))p_t(x)dx + \int_{0}^{\infty} \log(cw_0 + (1-c)(2-w_0))p_t(x)dx\right]$$

$$\overset{(ii)}{=} \underset{w_0}{\arg\min}\left[\log((1-c)w_0 + c(2-w_0))p_t(x \leq 0) + \log(cw_0 + (1-c)(2-w_0))p_t(x \geq 0)\right],$$

where equality (i) follows from $w_1 = 2 - w_0$ and the predictor function and (ii) follows from the fact that within each integral, the term inside the log is independent of $x$. Differentiating w.r.t. to $w_0$, we have:

$$\frac{1-2c}{2c+w_0-2cw_0}p_t(x \leq 0) + \frac{2c-1}{2cw_0+2-2c-w_0}p_t(x \geq 0) = 0$$

$$\frac{1}{2c+w_0-2cw_0}p_t(x \leq 0) + \frac{-1}{2cw_0+2-2c-w_0}(1 - p_t(x \leq 0)) = 0$$

$$(2cw_0+2-2c-w_0)p_t(x \leq 0) - (2c+w_0-2cw_0)(1 - p_t(x \leq 0)) = 0$$

$$2p_t(x \leq 0) - 2c - w_0 + 2cw_0 = 0,$$

which gives $w_0 = \frac{2p_t(x \leq 0) - 2c}{1 - 2c}$. Thus for the population MLLS estimate, the estimation error is given by

$$\|\widehat{w} - w^*\| = 2|w_0 - 2\alpha| = 4\left|\frac{(1-2\alpha)(p_s(x \geq 0|y=0) - c)}{1-2c}\right|.$$

$\square$

# E   Proofs from Section 5

The gradient of the MLLS objective can be written as

$$\nabla_w \mathcal{L}(w, f) = \mathbb{E}_t\left[\frac{f(x)}{f(x)^T w}\right], \tag{17}$$

and the Hessian is

$$\nabla_w^2 \mathcal{L}(w, f) = -\mathbb{E}_t\left[\frac{f(x)f(x)^T}{(f(x)^T w)^2}\right]. \tag{18}$$

We use $\lambda_{\min}(X)$ to denote the minimum eigenvalue of the matrix $X$.

**Lemma 9** (Theorem 5.1.1 [23]). *Let $X_1, X_2, \ldots, X_n$ be a finite sequence of identically distributed independent, random, symmetric matrices with common dimension $k$. Assume $0 \preceq X \preceq R \cdot I$ and $\mu_{\min}I \preceq \mathbb{E}[X] \preceq \mu_{\max}I$. With probability at least $1 - \delta$,*

$$\lambda_{\min}\left(\frac{1}{n}\sum_{i=1}^{n}X_i\right) \geq \mu_{\min} - \sqrt{\frac{2R\mu_{\min}\log(\frac{k}{\delta})}{n}}. \tag{19}$$

**Lemma 3.** *For any predictor $f$ that satisfies Condition 1, we have $\|w_f - \widehat{w}_f\| \leq \sigma_{f,w_f}^{-1}\mathcal{O}_p\left(m^{-1/2}\right)$.*

*Proof.* We present our proof in two steps. Step-1 is the non-probabilistic part, i.e., bounding the error $\|\widehat{w}_f - w_f\|$ in terms of the gradient difference $\|\nabla_w\mathcal{L}(w_f, f) - \nabla_w\mathcal{L}_m(w_f, f)\|$. This step uses Taylor's expansion upto second order terms for empirical log-likelihood around the true $w^*$. Step-2 involves deriving a concentration on the gradient difference using the Lipschitz property implied by Condition 1. Combining these two steps along with Lemma 22 concludes the proof. Now we detail each of these steps.

**Step-1.** We represent the empirical Negative Log-Likelihood (NLL) function with $\mathcal{L}_m$ by absorbing the negative sign to simplify notation. Using a Taylor expansion, we have

$$\mathcal{L}_m(\widehat{w}_f, f) = \mathcal{L}_m(w_f, f) + \langle \nabla_w \mathcal{L}_m(w_f, f), \widehat{w}_f - w_f \rangle + \frac{1}{2}(\widehat{w}_f - w_f)^T \nabla_w^2 \mathcal{L}_m(\widetilde{w}, f_c)(\widehat{w}_f - w_f),$$

where $\widetilde{w} \in [\widehat{w}_f, w_f]$. With the assumption $f^T w_f \geq \tau$, we have $\nabla_w^2 \mathcal{L}_m(\widetilde{w}, f) \geq \frac{\tau^2}{\min p_s(y)^2} \nabla_w^2 \mathcal{L}_m(w_f, f)$. Let $\kappa = \frac{\tau^2}{\min p_s(y)^2}$. Using this we get,

$$\mathcal{L}_m(\widehat{w}_f, f) \geq \mathcal{L}_m(w_f, f) + \langle \nabla_w \mathcal{L}_m(w_f, f), \widehat{w}_f - w_f \rangle + \frac{\kappa}{2}(\widehat{w}_f - w_f)^T \nabla_w^2 \mathcal{L}_m(w_f, f)(\widehat{w}_f - w_f)$$

$$\underbrace{\mathcal{L}_m(\widehat{w}_f, f) - \mathcal{L}_m(w_f, f)}_{\text{I}} - \langle \nabla_w \mathcal{L}_m(w_f, f), \widehat{w}_f - w_f \rangle \geq \frac{\kappa}{2}(\widehat{w}_f - w_f)^T \nabla_w^2 \mathcal{L}_m(w_f, f)(\widehat{w}_f - w_f),$$

where term-I is less than zero as $\widehat{w}_f$ is the minimizer of empirical NLL $\mathcal{L}_m(\widehat{w}_f, f)$. Ignoring term-I and re-arranging a few terms we get:

$$-\langle \nabla_w \mathcal{L}_m(w_f, f), \widehat{w}_f - w_f \rangle \geq \frac{\kappa}{2}(\widehat{w}_f - w_f)^T \nabla_w^2 \mathcal{L}_m(w_f, f)(\widehat{w}_f - w_f),$$

With first order optimality on $w_f$, $\langle \nabla_w \mathcal{L}(w_f, f), \widehat{w}_f - w_f \rangle \geq 0$. Plugging in this, we have,

$$\langle \nabla_w \mathcal{L}(w_f, f) - \nabla_w \mathcal{L}_m(w_f, f), \widehat{w}_f - w_f \rangle \geq \frac{\kappa}{2}(\widehat{w}_f - w_f)^T \nabla_w^2 \mathcal{L}_m(w_f, f)(\widehat{w}_f - w_f),$$

Using Holder's inequality on the LHS we have,

$$\|\nabla_w \mathcal{L}(w_f, f) - \nabla_w \mathcal{L}_m(w_f, f)\| \, \|\widehat{w}_f - w_f\| \geq \frac{\kappa}{2}(\widehat{w}_f - w_f)^T \nabla_w^2 \mathcal{L}_m(w_f, f)(\widehat{w}_f - w_f).$$

Let $\widehat{\sigma}_{f,w_f}$ be the minimum eigenvalue of $\nabla_w^2 \mathcal{L}_m(w^*, f_c)$. Using the fact that $(\widehat{w}_f - w_f)^T \nabla_w^2 \mathcal{L}_m(w_f, f)(\widehat{w}_f - w_f) \geq \widehat{\sigma}_{\min} \|\widehat{w}_f - w_f\|^2$, we get,

$$\|\nabla_w \mathcal{L}(w_f, f) - \nabla_w \mathcal{L}_m(w_f, f)\| \geq \frac{\kappa \widehat{\sigma}_{f,w_f}}{2} \|\widehat{w}_f - w_f\| . \tag{20}$$

**Step-2.** The empirical gradient is $\nabla_w \mathcal{L}_m(w_f, f) = \sum_{i=1}^m \frac{\nabla_w \mathcal{L}_1(x_i, w_f, f)}{m}$ where $\nabla \mathcal{L}_1(x_i, w_f, f) = \left[ \frac{f_1(x_i)}{\langle f(x_i), w_f \rangle} \cdots \frac{f_l(x_i)}{\langle f(x_i), w_f \rangle} \cdots \frac{f_k(x_i)}{\langle f(x_i), w_f \rangle} \right]_{(k)}$. With the lower bound $\tau$ on $f^T w_f$, we can upper bound the gradient terms as

$$\|\nabla_w \mathcal{L}_1(x, w_f, f)\| \leq \frac{\|f\|}{\tau} \leq \frac{\|f\|_1}{\tau} \leq \frac{1}{\tau} .$$

As the gradient terms decompose and are independent, using Hoeffding's inequality we have with probability at least $1 - \frac{\delta}{2}$,

$$\|\nabla_w \mathcal{L}(w_f, f) - \nabla_w \mathcal{L}_m(w_f, f)\| \leq \frac{1}{2\tau} \sqrt{\frac{\log(4/\delta)}{m}} . \tag{21}$$

Let $\sigma_{f,w_f}$ be the minimum eigenvalue of $\nabla_w^2 \mathcal{L}(w_f, f)$. Using lemma 9, with probability at least $1 - \frac{\delta}{2}$,

$$\frac{\widehat{\sigma}_{f,w_f}}{\sigma_{f,w_f}} \geq 1 - \tau \sqrt{\frac{\log(2k/\delta)}{m}} . \tag{22}$$

Plugging (21) and (22) in (20), and applying a union bound, we conclude that with probability at least $1 - \delta$,

$$\|\widehat{w}_f - w_f\|_2 \leq \frac{1}{\kappa \tau}\left( \sigma_{f,w_f} - \sigma_{f,w_f} \tau \sqrt{\frac{\log(2k/\delta)}{m}} \right)^{-1}\left( \sqrt{\frac{\log(4/\delta)}{m}} \right)$$

$$\leq \frac{1}{\kappa \tau} \frac{1}{\sigma_{f,w_f}}\left( 1 + \tau \sqrt{\frac{\log(2k/\delta)}{m}} \right) \sqrt{\frac{\log(4/\delta)}{m}} .$$

Neglecting the order $m$ term and letting $c = \frac{1}{\kappa \tau}$, we have

$$\|\widehat{w}_f - w_f\| \leq \frac{c}{\sigma_{f,w_f}} \sqrt{\frac{\log(4/\delta)}{m}} .$$

$\square$

**Lemma 4.** *For any predictor $f$ and any calibrated predictor $f_c$ that satisfies Condition 1, we have* $\|w_f - w^*\| \le \sigma_{f,w^*}^{-1} \cdot C \cdot \mathbb{E}_t\left[\|f - f_c\|\right]$ *,for some constant $C$.*

*If we set $f_c(x) = p_s(y|f(x))$, which is a calibrated predictor (Proposition 2), we can bound the error in terms of the calibration error of $f$ on the source data* [9]: $\|w_f - w^*\| \le \sigma_{f,w^*}^{-1} \cdot C \cdot \mathcal{E}(f)$.

*Proof.* We present our proof in two steps. Note, all calculations are non-probabilistic. Step-1 involves bounding the error $\|w_f - w^*\|$ in terms of the gradient difference $\|\nabla_w \mathcal{L}(w^*, f_c) - \nabla_w \mathcal{L}(w^*, f)\|$. This step uses Taylor's expansion on $\mathcal{L}(w_f, f)$ upto the second orderth term for population log-likelihood around the true $w^*$. Step-2 involves deriving a bound on the gradient difference in terms of the difference $\|f - f_c\|$ using the Lipschitz property implied by Condition 1. Further, for a crude calibration choice of $f_c(x) = p_s(\cdot|x)$, the gradient difference can be bounded by miscalibration error. We now detail both of these steps.

**Step-1.** Similar to Lemma 3, we represent with $\mathcal{L}$ by absorbing the negative sign to simplify notation. Using the Taylor expansion, we have

$$\mathcal{L}(w_f, f) \ge \mathcal{L}(w^*, f) + \langle \nabla_w \mathcal{L}(w^*, f), w_f - w^* \rangle + \frac{1}{2}(w_f - w^*)^T \nabla_w^2 \mathcal{L}(\widetilde{w}, f)(w_f - w^*),$$

where $\widetilde{w} \in [w_f, w^*]$. With the assumption $f^T w^* \ge \tau$, we have $\nabla_w^2 \mathcal{L}(\widetilde{w}, f) \ge \frac{\tau^2}{\min p_s(y)^2} \nabla_w^2 \mathcal{L}(w^*, f)$. Let $\kappa = \frac{\tau^2}{\min p_s(y)^2}$. Using this we get,

$$\mathcal{L}(w_f, f) \ge \mathcal{L}(w^*, f) + \langle \nabla_w \mathcal{L}(w^*, f), w_f - w^* \rangle + \frac{\kappa}{2}(w_f - w^*)^T \nabla_w^2 \mathcal{L}(w^*, f)(w_f - w^*)$$

$$\underbrace{\mathcal{L}(w_f, f) - \mathcal{L}(w^*, f)}_{\text{I}} \ge \langle \nabla_w \mathcal{L}(w_f, f), w_f - w^* \rangle + \frac{\kappa}{2}(w_f - w^*)^T \nabla_w^2 \mathcal{L}(w^*, f)(w_f - w^*),$$

where term-I is less than zero as $w_f$ is the minimizer of NLL $\mathcal{L}(w, f)$. Ignoring that term and re-arranging a few terms we get

$$-\langle \nabla_w \mathcal{L}(w^*, f), w_f - w^* \rangle \ge \frac{\kappa}{2}(w_f - w^*)^T \nabla_w^2 \mathcal{L}(w^*, f)(w_f - w^*).$$

With first order optimality on $w^*$, $\langle \nabla_w \mathcal{L}(w^*, f_c), w_f - w^* \rangle \ge 0$. Using this we have:

$$\langle \nabla_w \mathcal{L}(w^*, f_c), w_f - w^* \rangle - \langle \nabla_w \mathcal{L}(w^*, f), w_f - w^* \rangle \ge \frac{\kappa}{2}(w_f - w^*)^T \nabla_w^2 \mathcal{L}(w^*, f)(w_f - w^*),$$

$$\langle \nabla_w \mathcal{L}(w^*, f_c) - \nabla_w \mathcal{L}(w^*, f), w_f - w^* \rangle \ge \frac{\kappa}{2}(w_f - w^*)^T \nabla_w^2 \mathcal{L}(w^*, f)(w_f - w^*).$$

As before, let $\sigma_{f,w}$ be the minimum eigenvalue of $\nabla_w^2 \mathcal{L}(w^*, f)$. Using the fact that $(w_f - w^*)^T \nabla_w^2 \mathcal{L}(w^*, f)(w_f - w^*) \ge \sigma_{f,w} \|w_f - w^*\|^2$, we get

$$\langle \nabla_w \mathcal{L}(w^*, f_c) - \nabla_w \mathcal{L}(w^*, f), w_f - w^* \rangle \ge \frac{\kappa \sigma_{f,w}}{2} \|w_f - w^*\|^2.$$

Using Holder's inequality on the LHS and re-arranging terms gives

$$\|\nabla_w \mathcal{L}(w^*, f_c) - \nabla_w \mathcal{L}(w^*, f)\| \ge \frac{\kappa \sigma_{f,w}}{2} \|w_f - w^*\|. \tag{23}$$

**Step-2.** By lower bound assumptions $f_c^T w^* \ge \tau$ and $f^T w^* \ge \tau$, we have

$$\|\nabla_w \mathcal{L}(w^*, f_c) - \nabla \mathcal{L}(w^*, f)\| \le \mathbb{E}_t\left[\|\nabla \mathcal{L}_1(x, w^*, f_c) - \nabla \mathcal{L}_1(x, w^*, f)\|\right] \le \frac{1}{\tau^2} \mathbb{E}_t\left[\|f_c(x) - f(x)\|\right], \tag{24}$$

where the first inequality is implied by Jensen's inequality and the second is implied by the Lipschitz property of the gradient. Further, we have

$$\mathbb{E}_t\left[\|f_c(x) - f(x)\|\right] = \mathbb{E}_s\left[\frac{p_t(x)}{p_s(x)} \|f_c(x) - f(x)\|\right]$$

$$\leq \mathbb{E}_s \left[ \max_y \frac{p_t(y)}{p_s(y)} \| f_c(x) - f(x) \| \right]$$

$$\leq \max_y \frac{p_t(y)}{p_s(y)} \mathbb{E}_s \left[ \| f_c(x) - f(x) \| \right] . \tag{25}$$

Combining equations (23), (24), and (25), we have

$$\| w_f - w^* \| \leq \frac{2}{\kappa \sigma_{f,w} \tau^2} \max_y \frac{p_t(y)}{p_s(y)} \mathbb{E}_s \left[ \| f_c(x) - f(x) \| \right] . \tag{26}$$

Further, if we set $f_c(x) = p_s(\cdot | f(x))$, which is a calibrated predictor according to Proposition 2, we can bound the error on the RHS in terms of the calibration error of $f$. Moreover, in the label shift estimation problem, we have the assumption that $p_s(y) \geq c > 0$ for all $y$. Hence, we have $\max_y p_t(y)/p_s(y) \leq 1/c$. Using Jensen's inequality, we get

$$\mathbb{E}_s \| f_c(x) - f(x) \| \leq \left( \mathbb{E}_s \| f_c(x) - f(x) \|^2 \right)^{\frac{1}{2}} = \mathcal{E}(f) . \tag{27}$$

Plugging (27) back in (26), we get the required upper bound. $\qquad \square$

**Proposition 3.** *For any $w \in \mathcal{W}$, we have $\sigma_{f,w} \geq p_{s,\min} \sigma_f$ where $\sigma_f$ is the minimum eigenvalue of $\mathbb{E}_t \left[ f(x) f(x)^T \right]$ and $p_{s,\min} = \min_{y \in \mathcal{Y}} p_s(y)$. Furthermore, if $f$ satisfies Condition 1, we have $p_{s,\min}^2 \cdot \sigma_f \leq \sigma_{f,w} \leq \tau^{-2} \cdot \sigma_f$, for $w \in \{ w_f, w^* \}$.*

*Proof.* For any $v \in \mathbb{R}^k$, we have

$$v^T \left( -\nabla_w^2 \mathcal{L}(w, f) \right) v = \mathbb{E}_t \left[ \frac{\left( v^T f(x) \right)^2}{\left( f(x)^T w \right)^2} \right] \in \left[ \frac{1}{a^2}, \frac{1}{b^2} \right] \cdot v^T \mathbb{E}_t \left[ f(x) f(x)^T \right] v ,$$

where

$$a = \max_{x : p_s(x) > 0} f(x)^T w \leq \frac{1}{p_{s,\min}}$$

and

$$b = \min_{x : p_s(x) > 0} f(x)^T w \geq \tau$$

if $f$ satisfies Condition 1 and $w \in \{ w_f, w^* \}$. Therefore, we have

$$p_{s,\min}^2 \cdot \sigma_f \leq \sigma_{f,w} \leq \tau^{-2} \cdot \sigma_f$$

for $w \in \{ w_f, w^* \}$.

$\qquad \square$

**Lemma 5.** *Let $f = g \circ \hat{f}$ be the predictor after post-hoc calibration with squared loss $l$ and $g$ belongs to a function class $\mathcal{G}$ that satisfies the standard regularity conditions, we have*

$$\mathcal{E}(f) \leq \min_{g \in \mathcal{G}} \mathcal{E}(g \circ \hat{f}) + \mathcal{O}_p \left( n^{-1/2} \right) . \tag{8}$$

*Proof.* Assume regularity conditions on the model class $\mathcal{G}_\theta$ (injectivity, Lipschitz-continuity, twice differentiability, non-singular Hessian, and consistency) as in Theorem 5.23 of Stein [21] hold true. Using the injectivity property of the model class as in Kumar et al. [14], we have for all $g_1, g_2 \in \mathcal{G}$,

$$\text{MSE}(g_1) - \text{MSE}(g_2) = \mathcal{E}(g_1)^2 - \mathcal{E}(g_2)^2 . \tag{28}$$

Let $\hat{g}, g^* \in \mathcal{G}$ be models parameterized by $\hat{\theta}$ and $\theta^*$, respectively. Using the strong convexity of the empirical mean squared error we have,

$$\text{MSE}_n(\hat{g}) \geq \text{MSE}_n(g^*) + \langle \nabla_\theta \text{MSE}_n(g^*), \hat{\theta} - \theta^* \rangle + \frac{\mu^2}{2} \left\| \hat{\theta} - \theta^* \right\|_2^2 ,$$

where $\mu$ is the parameter constant for strong convexity. Re-arranging a few terms, we have

$$\underbrace{\text{MSE}_n(\widehat{g}) - \text{MSE}_n(g^*)}_{\text{I}} - \langle \nabla_\theta \text{MSE}_n(g^*), \widehat{\theta} - \theta^* \rangle \geq \frac{\mu^2}{2} \|\widehat{\theta} - \theta^*\|_2^2 \,,$$

where term-I is less than zero because $\widehat{g}$ is the empirical minimizer of the mean-squared error. Ignoring term-I, we get:

$$\frac{\mu^2}{2} \|\widehat{\theta} - \theta^*\|_2^2 \leq -\langle \nabla_\theta \text{MSE}_n(g^*), \widehat{\theta} - \theta^* \rangle \leq \|\nabla_\theta \text{MSE}_n(g^*)\| \left\|\widehat{\theta} - \theta^*\right\| \,.$$

As the assumed model class is Lipschitz w.r.t. $\theta$, the gradient is bounded by Lipschitz constant $L = c_1$. $\mathbb{E}\left[\nabla_\theta \text{MSE}_n(g^*)\right] = 0$ as $g^*$ is the population minimizer. Using Hoeffding's bound for bounded functions, we have with probability at least $1 - \delta$,

$$\|\widehat{\theta} - \theta^*\|_2 \leq \frac{c_1}{\mu^2} \sqrt{\frac{\log(2/\delta)}{n}} \,. \tag{29}$$

Using the smoothness of the $\text{MSE}(g)$, we have

$$\text{MSE}(\widehat{g}) - \text{MSE}(g^*) \leq c_2 \|\widehat{\theta} - \theta^*\|_2^2 \,, \tag{30}$$

where $c_2$ is the operator norm of the $\nabla^2 \text{MSE}(g^*)$. Combining (28), (29), and (30), we have for some universal constant $c = \frac{c_1 c_2}{\mu^2}$ with probability at least $1 - \delta$,

$$\mathcal{E}(\widehat{g})^2 - \mathcal{E}(g^*)^2 \leq c \frac{\log(2/\delta)}{n} \,.$$

$\square$

Moreover, with Lemma 4, depending on the degree of the miscalibration and the method involved to calibrate, we can bound the $\mathcal{E}(f)$. For example, if using vector scaling on a held out training data for calibration, we can use Lemma 5 to bound the calibration error $\mathcal{E}(f)$, i.e., with probability at least $1 - \delta$, we have

$$\mathcal{E}(f) \leq \sqrt{\min_{g \in \mathcal{G}} \mathcal{E}(g \circ f)^2 + c \frac{\log(2/\delta)}{n}} \leq \min_{g \in \mathcal{G}} \mathcal{E}(g \circ f) + \sqrt{c \frac{\log(2/\delta)}{n}} \,. \tag{31}$$

Plugging (27) and (31) into (26), we have with probability at least $1 - \delta$ that

$$\|w_f - w^*\| \leq \frac{1}{\kappa \sigma_{f,w} \tau^2} \left( \|w^*\|_2 \left( \sqrt{c \frac{\log(2/\delta)}{n}} + \min_{g \in \mathcal{G}} \mathcal{E}(g \circ f) \right) \right) \,.$$

## Footnotes

[7]Motivated by the strong empirical results in Alexandari et al. [1], we use BCTS in our experiments as a surrogate for canonical calibration.

[8]Note that we use the term *density* loosely here for convenience. The actual density is $\langle f(x), w\rangle \cdot p_s(x)$ but we can ignore $p_s(x)$ because it does not depend on our parameters.

[9]We present two upper bounds because the second is more interpretable while the first is tighter.