[Reviews · NeurIPS 2020]

Review 1

Summary and Contributions: The paper takes the approaches commonly which commonly utilizes black box predictors to estimate label shift and then proposes a common framework through which one can analyze them. The paper, then proceeds to prove the conditions of consistency for these estimators, given estimation error analysis of the two methods (BBSE and MLLS) and shows why BBSE methods under perform as compared MLLS method.

Strengths: The paper casts the MLLS with a calibrated predictor as a distribution matching problem and notices that BBSE method can also be casted in the same way, thereby giving a unified view. The theoretical analysis of the consistency of these estimators looks sound. The paper proposes an explanation of why BBSE methods might be doing worse than MLLS and then proceeds to give empirical justification for the same. Prior work like RLLS give finite sample error bounds on the estimate. However, this work additionally shows that, it is not just the finite sample error that is in play, there is a role of how accurate the calibration is, which also plays a role in the final estimation error, which makes sense.

Weaknesses: The work currently restricts the analysis to black box predictors. However, it might have been possible to analyze even white box predictors, by assuming some structure on f.

Correctness: The claims made in the paper look correct, the proof of the theoretical analysis looks correct. The experimentations done are mainly to support some of the theoretical claims made and they look correct too.

Clarity: The title feels a little broad compared to the work. The work is focused on label shift estimation using black box predictors. It would be good to modify the title to reflect this scoped problem. Otherwise, the paper is well written and easy to understand.

Relation to Prior Work: There are no prior works that theoretically analyze MLLS and also provide a common frame work for both MLLS and BBSE estimators. The paper does, however, give prior works like RLLS which is a BBSE estimator and shows why MLLS methods may do better than such estimators.

Reproducibility: Yes

Additional Feedback: In the code, there are some hardcoded paths - which may not hold true on all machines. The README does not give instructions on how this paths need to be updated. Execution of the code takes a long time with no print statement, making it unclear as to what is happening. Some idea of what kind of machines were used to execute this code and what is the expected time to run the code would have been useful. ---- Post Author Response: Thanks to the authors for their response. I have read through the other reviews and also the responses of the authors. After going through them, I am still feeling quite positive about this work and will retain my earlier evaluation of acceptance of the paper.


Review 2

Summary and Contributions: In this paper, authors provided a unifying framework which summarizes two major approaches in label shift estimation: BBSE and MLLS. The consistency and estimation error are proved for MLLS. The result implies that BBSE's inefficiency is probably due to a loss of information when calibrating the confusion matrix.

Strengths: The first strength of this paper is it summarizes MLLS and BBSE using a distribution matching framework. To my best knowledge, this is the first time such a framework has been proposed, although it is not a hugly surprising result given BBSE and MLLS are both based on label shift assumption (p_sX|y = p_tX|y). However, this derivation allows authors later to derive the consistency for MLLS using classic maximum likelihood results. The second contribution is that the estimation error of MLLS of can be decomposed into finite sample approximation error and miscalibration error. However I am not sure how the calibration error can be determined in practice which would be really useful. I imagine this is something like an oracle inequality that cannot be quantified? It is also important that authors have pointed out that the calibratedness of the predictor f as a sufficient and necessary condition to the consistency of MLLS which aligns with previous empirical observations. The unified framework gives rise to MLLS-CS which has a similar performance to BBSE. This is a very interesting result as it shows the differnece in terms of perforamnce is due to the aggregation of confusion matrix. This is a very important mesage and is not knwon to the community to my best knowledge.

Weaknesses: I do not find major flaws in this paper. A few minor points: 1. Should line 165 be E_t [log p_s(z)] not E_t [p_s(z)] ? 2. Figure 2 is hard to read: which lines correspond to the left scale and which lines correspond to the right scale? 3. Line 323, Figure (6) -> Figure 1?

Correctness: The method and claims seems sound.

Clarity: Thisa paper is well-written and I do not have troubles following the main ideas of this paper.

Relation to Prior Work: This paper is not directly related to previous works and other related works are properly referenced.

Reproducibility: Yes

Additional Feedback: Thanks authors for replying our comments. I do not have further questions.


Review 3

Summary and Contributions: This paper studies the label estimation problem and unifies a previously proposed perspective--maximum likelihood and calibration-- with the recent method using black-box predictors. The main takeaway is that these two perspectives can be regarded as the same framework and the calibration is a necessary step to achieve better performance. In the analysis, the weight estimation error is analyzed by decomposing it into an estimation error due to finite samples and an calibration error due to label shift. The empirical evaluation demonstrates that the combined method (MLLS with confusion matrix) outperforms using only black-box predictors. ======= Thanks for the author response. I keep my score after reading it. Some of my concerns are resolved. The following to remains: The first is about the "miscalibration error", which I actually mean the gap between target error (the RHS in lemma4 part 1) and the calibrated source error (the RHS in lemma4 part 2). Intuitively, the larger the source and target difference are, the larger the gap is. Eventually, the bound is presented using calibrated source error, which can be very loose. The second is about weight estimation and downstream tasks. It is true that most previous analysis combines weight estimation analysis with a standard IW learning algorithm analysis and obtain similar guarantees as lemma 5. However, it is not validated empirically in this paper that the proposed method would benefit the downstream tasks. Also, even in previous work, there are practical cases when this reflection in the improvement of learning is not obvious. I encourage the authors to further improve the paper.

Strengths: The label shift estimation problem is very relevant to the community. The perspective of the unification of two perspectives in label shift estimation is novel and interesting. The derivation and analysis in this paper is sound and clear.

Weaknesses: The proposed method is not very novel, since it builds on the previous two perspectives. So if looking at the specific steps for the algorithm, they are not new to the community. However, given that, I still think the paper would benefit from having an algorithm to better summarize the proposed method. The analysis of the miscalibration error is not very informative. The miscalibration error can be quite large if the source and target data differ a lot. So it seems this analysis does not support the argument that the calibration is the main reason that MLLS outperforms methods using black-box predictor. A good label shift estimation does not mean it would perform well in shift correction or other down-stream tasks. This paper only focuses on half of the problem while in reality, it is the end goal (adaptation/learning with the weights) that matters.

Correctness: The technical content is mostly correct. Regarding the calibration and the performance I have the following concerns: This paper uses empirical results and a synthetic example to show that the performance gain of MLLS is due to the calibration, and sometimes it is necessary. This is a reasonable claim but there is still a gap in this question, which is when and why. Since the calibration in reality is not perfect, then the questions are when it is necessary and why in practice, it seems to be always better. Does it have anything to do with the label shift magnitude? Can we do a careful ablation study to see when calibration using source data would fail? A related point is also that BCTS-calibrated method is dominating in the experiments, which is not the focus of discussion in the paper. This mismatch between experiments and theoretical analysis make me feel that the claims and insights from the analysis is not validated in the experiments.

Clarity: The paper is generally well-written.

Relation to Prior Work: This paper surveys related work extensively and discusses connections and differences with them clearly.

Reproducibility: Yes

Additional Feedback:


Review 4

Summary and Contributions: The authors explore a number of prior arts in supervised learning under label shift with a focus on Black Box Shift Estimation (BBSE) and Maximum Likelihood Label Shift (MLLS). They show how calibration is an essential step to obtain good properties for MLLS. They show that both BBSE and MLLS can be regarded as two instances of a more general class of label shift correction mechanisms that rely on (i) designing a latent space for calibration and (ii) use a distribution matching technique in this latent space. Specifically for MLSS, the authors also provide error bounds to take into account the finiteness of the sample which induces an estimation error of the parameters of the latent space distribution family and in the calibration procedure.

Strengths: The paper is theoretically well grounded and addresses an interesting practical ML problem. The claims are supported by empirical evidence. References and benchmarked methods are up to date and the novelty of the contribution with respect to them is clearly stated.

Weaknesses: In my opinion, the main weakness of the paper is that the addressed problem is one of the simplest form of data generative distribution mismatch between train and test times. The label shift assumption is a strong assumption which in practice may not hold. The authors do not discuss how the label shift correction framework they established could be plugged into more general domain adaptation techniques.

Correctness: I have checked some of the proofs in details and found them correct. The empirical methodology is adapted to the tackled issue.

Clarity: The paper reads pretty well. In my opinion, there is only one odd wording: I suggest to replace "absent of .." by "in the absence of .."

Relation to Prior Work: The relation to prior art is clear as the paper is meant to bind some of these arts under a common framework and draw new conclusions on their respective abilities to address label shift correction.

Reproducibility: Yes

Additional Feedback: Even if I would have liked to see some perspectives on more general forms of domain adaption, I believe the paper is insightful to the ML community and contains interesting take-home-messages. ---- Post Author Response: the authors have provided a fair feedback to some of my remarks. After reading their answers and the other reviews, I have the feeling the paper qualities outweigh its shortcomings. I therefore maintain my score.

[Author Response · NeurIPS 2020]

We would like to thank the reviewers for thoughtful feedback. We are glad to see that all 4 reviewers vote for acceptance,
with the reviewers recognizing the paper to be "insightful", "novel and interesting", and to have "very important
take-home-messages". Overall, the reviewers appreciated the unified view framework (R1, R3, R4, R5), our theory's
relevance (R1, R3, R5), and our extensive experiments (R1, R3, R5). Below, we respond to each reviewer's concerns:

**Reply to R1:** Thank you for the positive assessment and for highlighting the importance of our theoretical results.

**"analyze even white box predictors":** This is an interesting suggestion. Perhaps by leveraging the structure of a
predictor, one might be able to design more efficient estimators.

**"The title feels a little broad compared to the work."** Thanks for this feedback. We will review all claims, including
the title, for the camera-ready version to make sure that our contributions are presented accurately.

**"In the code, there are some hardcoded paths..."** Sorry for this inconvenience. For the camera-ready, we will fix the
hardcoded paths and provide detailed instructions for execution with expected run time.

**Reply to R3:** Thank you for the positive feedback and for championing our paper. We are glad that you found our
message to be very important and unknown to the community. We will fix Figure 2 and typos in the final.

**"how the calibration error can be determined in practice."** Procedure for estimating canonical calibration error
from samples do not yet exist. However, surrogate measures like Expected Calibration Error work well in practice.
Estimating canonical calibration error and efficiently calibrating models remain important open problems.

**Reply to R4:** Thanks for the detailed review and positive assessment. We are glad that you appreciate the novelty of
our unified framework and consider the label shift problem relevant to the community.

**"The proposed method is not very novel."** The novelty of our contribution does not include the introduction of new
estimation methods. Instead, our work contributes the theoretical underpinnings for understanding popular methods.
We believe that this form of novelty is equally deserving of publication as the introduction as new methods.

**"The analysis of the miscalibration error is not very informative.** Our theory shows two-fold benefits of calibration:
(i) canonical calibration (Def 1) and an invertible confusion matrix (as BBSE requires) are necessary & sufficient for
MLLS's consistency (Th 1-2, Cor 1). (ii) We bound one term in the finite sample error by miscalibration error (Lem 4).

**"The miscalibration error can be large if the source and target data differ."** Our bounds hold with calibration error
*on the source distribution only*. While the RHS in the first part of Lemma 4 is unobservable, it is upper-bound by the
calibration error on the source domain (the second part of Lemma 4). The multiplicative constant in the bounding step
depends on $\max_y p_t(y)/p_s(y)(>0)$ which also appears in existing blackbox estimation guarantees (e.g. BBSE, RLLS).

**"analysis does not support that the calibration is the main reason that MLLS outperforms blackbox methods."**
Our finite-sample error bound hint at the importance of calibration (Th 3). Informally, the bound highlights dependency
on 2 factors: (i) calibration error on source data; (ii) minimum eigenvalue of the Hessian of the likelihood (which
increases with the granularity of calibration). The first factor explains the superior performance of calibrated MLLS
over uncalibrated MLLS and is consistent with existing empirical observations (L 177-180). The latter factor elucidates
the efficacy of BCTS-calibrated MLLS over BBSE. When MLLS is calibrated with BCTS on source data (Lem 5, L
295-298), granular calibration in practice tightens the bound (L 299-307).

**"A good label shift estimation does not mean it would perform well in down-stream tasks."** Existing bounds
on downstream classification error (under label shift) depend on the MSE of the label distribution estimates (Th 1
Azizzadenesheli 2019[2])—thus improving these estimates translates into better downstream guarantees (Lines 83-85).

**"but there is still a gap in this question, which is when and why (calibration is necessary)."** Calibration on the
source distribution is sufficient (Th 2). Weaker notions of calibration (e.g. marginal calibration) is insufficient to
guarantee consistency (L 228-230). We agree that we only proved sufficiency and haven't fully characterize the necessity
of calibration. Necessity is shown for only a restrict class of classifiers (e.g. the thresholding classifiers in Ex 1).

**"BCTS method is dominating in the experiments. This mismatch is not validated in the experiments."** Inspired
from Alexendari 2020[1], we use BCTS as a surrogate for canonical calibration (no known method is guaranteed to
achieve canonical calibration). However, in our synthetic GMM setting, we can perfectly calibrate the classifier (without
BCTS) and show clearly how more granular calibration improves MLLS estimates (Figure 2, Lines 337-344).

**Reply to R5:** We thank the reviewer for the thoughtful review and positive feedback. We are glad that you consider our
work as insightful to the ML community with interesting take-home-messages.

**"Label shift is a strong assumption which in practice may not hold... how this can be plugged into more general
domain adaptation."** We agree that the label shift assumption is strong & unlikely to hold exactly in practice. However,
we believe that rigorously understanding these idealized settings is a fundamental building block towards more complex
settings. Moreover, in many important problems (e.g. medical diagnosis during an epidemic), label shift is nevertheless
a useful model because prevalence is likely to change faster than conditional probabilities of symptoms given disease.

[Meta-Review · NeurIPS 2020]

The reviewers concur to find the paper interesting, well written, and with compelling experiments. They made constructive comments which the authors are encouraged to consider when preparing the final version of the manuscript.